# Single-point mutated lanmodulin as a high-performance MRI contrast agent for vascular and kidney imaging

Yuxia Liu ®[1,2,11], Duyang Gao[3,4,11], Yuanyuan He[5,6], Jing Ma[5,6], Suet Yen Chong[7,8,9], Xinyi Qi[7,8,9], Hui Jun Ting[7,8,9], Zichao Luo ®[1,2], Zhigao Yi ®[1,2], Jingyu Tang[1,2], Chao Chang ®[5,6], Jiongwei Wang ®[7,8,9,10], Zonghai Sheng ®[3,4] ✉, Hairong Zheng ®[3,4] ✉ & Xiaogang Liu ®[1,2,7] ✉

Magnetic resonance imaging contrast agents can enhance diagnostic precision but often face limitations such as short imaging windows, low tissue specificity, suboptimal contrast enhancement, or potential toxicity, which affect resolution and long-term monitoring. Here, we present a protein contrast agent based on lanmodulin, engineered with a single-point mutation at position 108 from N to D to yield maximum gadolinium binding sites. After loading with $Gd^{3+}$ ions, the resulting protein complex, LanND-Gd, exhibits efficient renal clearance, high relaxivity, and prolonged renal retention compared to commercial agents. LanND-Gd enables high-performance visualization of whole-body structures and brain vasculature in male mice at a resolution finer than one hundred micrometers. In male ischemia mouse models, LanND-Gd also improves kidney dysfunction monitoring while minimizing risks of neural toxicity or immunogenic reactions. This protein-based contrast agent offers superior image quality, improved biocompatibility, and extended imaging timeframes, promising significant advancements in magnetic resonance-based diagnostics and patient outcomes.

The pursuit of high-performance clinical imaging modalities can largely improve diagnostic precision and patient care. In modern medical imaging, magnetic resonance imaging (MRI) is widely used to provide exquisite soft tissue contrast without using ionizing radiation. MRI contrast agents, especially those based on paramagnetic gadolinium ($Gd^{3+}$), are commonly employed to shorten the longitudinal relaxation time ($T_1$) of water protons, thereby enhancing MRI signal intensity[1].

Currently, all clinically approved $Gd^{3+}$-containing contrast agents are formulated with small chelators, resulting in low longitudinal relaxivities ($r_1$) in the range of 4–6 mM$^{-1}$ s$^{-1}$ due to their low water coordination number. This limitation in relaxivity largely constrains the sensitivity of existing MRI techniques, leading to inadequate resolution for visualizing complex biological structures, such as intricate brain vasculature at the micrometer scale. Such high-resolution imaging is

[1]Department of Chemistry, National University of Singapore, Singapore, Singapore. [2]The N.1 Institute for Health, National University of Singapore, Singapore, Singapore. [3]Biomedical Imaging Science and System Key Laboratory, Chinese Academy of Sciences, Shenzhen, China. [4]Paul C. Lauterbur Research Center for Biomedical Imaging, Institute of Biomedical and Health Engineering, Shenzhen Institute of Advanced Technology, Chinese Academy of Sciences, Shenzhen, China. [5]School of Physics, Peking University, Beijing, China. [6]Innovation Laboratory of Terahertz Biophysics, National Innovation Institute of Defense Technology, Beijing, China. [7]Department of Surgery, Yong Loo Lin School of Medicine, National University of Singapore, Singapore, Singapore. [8]Cardiovascular Research Institute, Yong Loo Lin School of Medicine, National University of Singapore, Singapore, Singapore. [9]Nanomedicine Translational Research Programme, Centre for NanoMedicine, Yong Loo Lin School of Medicine, National University of Singapore, Singapore, Singapore. [10]Department of Physiology, National University of Singapore, Singapore, Singapore. [11]These authors contributed equally: Yuxia Liu, Duyang Gao. ✉e-mail: zh.sheng@siat.ac.cn; hr.zheng@siat.ac.cn; chmlx@nus.edu.sg

crucial for the early diagnosis and prompt detection of cerebral diseases[1–5]. In addition, existing contrast agents have notably short retention time in the body, limiting the effective imaging window to less than 30 min[1]. This short duration creates challenges in real-time monitoring of dynamic functional processes, such as kidney disorders, where early diagnosis is imperative to prevent the progression to chronic kidney disease or renal failure[6]. Moreover, several noninvasive imaging techniques, including single-photon emission computed tomography, positron emission tomography, and computed tomography, involve the risk of radiation exposure[7]. Although emerging fluorescence imaging techniques are promising, their applications in clinical trials are hampered by their limited tissue penetration and interference caused by the natural fluorescence of tissues[8–10]. While nanoparticle-based contrast agents have been extensively explored to overcome the above issues, their unclear elimination routes hinder clinical translations[11–16]. Hence, there is an unmet need for developing an advanced MRI contrast agent for noninvasive and radiation-free visualization of fine vasculatures and clinical evaluation of organ dysfunction. The discovery of lanmodulin (LanM)[17–19], a high-affinity lanthanide-binding protein, has inspired the development of a high-performance MRI contrast agent due to its ability to coordinate two water molecules[20]. This unique property may potentially improve relaxivity performance compared to conventional clinical agents, which typically coordinate only one water molecule[21].

In this work, we engineer a LanM-derived protein MRI contrast agent for high-resolution vascular imaging and effective monitoring of kidney dysfunction (Fig. 1a). Through sequence alignment and structural analysis, we introduce a single-point mutation into the fourth weak ion-binding site of LanM, creating LanND, which fully saturates $Gd^{3+}$ binding and increases ion affinity, thereby reducing toxicity risks from ion leakage. After $Gd^{3+}$ loading, the LanND-Gd complex achieves a molecular relaxivity exceeding $50\,mM^{-1}\,s^{-1}$ at 3 T, which surpasses commercial agents ($<5\,mM^{-1}\,s^{-1}$) and enables high-resolution imaging of fine brain vasculature (up to $100\,\mu m$). Its extended renal retention also supports monitoring kidney dysfunction from ischemia. With a clear renal excretion pathway, high biocompatibility, and minimal neural toxicity, LanND-Gd presents a safe and high-performance MRI contrast agent, expanding the clinical potential of MRI applications.

## Results

### Design and characterization of LanM-derived $Gd^{3+}$ carriers
The original LanM contains a signaling peptide and four regions with enriched negatively-charged residues that enable cation binding (Supplementary Fig. 1). To increase protein yield, we truncated the N-terminal signaling peptide[17]. The LanM structure displays an EF-hand architecture similar to calmodulin (CaM)[22,23], with the first three EF-hand domains exhibiting strong $Ln^{3+}$ ion affinity (picomolar level) and the fourth domain having lower affinity (millimolar level) and forming a pseudo-binding site[22] (Fig. 1b). This condition raises a big concern for in vivo applications due to that the weakly-bound ion within the fourth binding pocket may easily dissociate from the protein to potentially cause toxicity. Upon aligning the sequences of EF-hand domains from both LanM and CaM, five negatively charged positions are highly conserved to coordinate cations (Fig. 1c). Therefore, we strategically introduced a single-point mutation, N108D, into the fourth EF-hand motif to create the mutant LanND, designed to modulate $Gd^{3+}$ binding stoichiometry. Structural predictions indicate that LanND preserves the spatial integrity of cation-binding pockets and increases surface negative charge density (Supplementary Fig. 2).

We subsequently purified both wild-type LanM and mutant LanND, as evident from clear SDS-PAGE gel bands (Fig. 1d and Supplementary Fig. 3). We first evaluated their secondary structural stability by circular dichroism (CD) spectroscopy. The CD spectra of the proteins showed bisignate patterns with positive signals below 200 nm and negative signals above 200 nm (Fig. 1e). Notably, these CD spectra shared consistent characteristics across various temperatures and pH levels, implying that the proteins maintain a stable helical structure under diverse conditions. We subsequently assessed their $Gd^{3+}$ binding capabilities using xylenol orange as an indicator for free $Gd^{3+}$ ions, revealing that single-point mutation in LanND increased its $Gd^{3+}$ binding sites compared to LanM. Specifically, the equivalent $Gd^{3+}$ stoichiometry increased from approximately 2.2 to 3.8 (Fig. 1f). The saturated binding stoichiometry is crucial, as it is difficult to completely remove all weakly bound $Gd^{3+}$ ions from EF4 in LanM using desalting columns and spin filters. This difficulty is evidenced by the bound $Gd^{3+}$ to LanM ratio exceeding 3, indicating that the wild-type LanM-Gd complex may be a risky choice for use as a contrast agent (Supplementary Fig. 4). Furthermore, the binding affinity of LanND was compared to LanM by dye-competition assays[24]. With the incremental addition of LanM or LanND, the fluorescence of a pre-prepared system containing equimolar amounts of Fluo-5N (a $Gd^{3+}$ indicator) and $Gd^{3+}$ gradually decreased (Fig. 1g). The effective binding affinity of LanND was determined as approximately $1.4 \times 10^{-12}\,M$, contrasting with $6 \times 10^{-12}\,M$ of LanM (Fig. 1h and Supplementary Fig. 5). This finding supports that the single-point mutation enhances the protein's metal-binding capability, showcasing its greater potential as a contrast agent. Importantly, the mutant did not induce abnormal aggregation or degradation in the presence or absence of $Gd^{3+}$ ions (Supplementary Fig. 6).

### MRI performance with the LanND-Gd complex
We introduced $Gd^{3+}$ to LanND and subsequently removed all unbound free ions through buffer exchange with desalting columns and protein spin filters, resulting in the protein-based contrast agent LanND-Gd (Fig. 1i). The filtrate obtained from the final centrifugation was collected for $Gd^{3+}$ analysis by inductively coupled plasma optical emission spectroscopy (ICP-OES) to confirm the complete removal of free $Gd^{3+}$. At a magnetic field strength of 3 T, LanND-Gd exhibited a brighter $T_1$-weighted image compared with the clinically used Magnevist (Fig. 1j). The average $T_1$ relaxivity ($r_1$) of LanND-Gd is $13.15\,mM^{-1}\,s^{-1}$, which significantly exceeds that of Magnevist with a relaxivity of $4.62\,mM^{-1}\,s^{-1}$ (Fig. 1k). Enhanced relaxivity was also observed in LanM-Gd, with comparable relaxivity values to LanND-Gd, indicating that the mutation does not perturb the protein's ability to coordinate two water molecules (Supplementary Fig. 7). The increased relaxivity of LanND-Gd is maintained at 7 T, consistent with values reported for LanM-Gd in other studies[25] (Supplementary Fig. 8). Notably, a single LanND-Gd molecule contains four $Gd^{3+}$-binding sites, while Magnevist has only one. As a result, the effective molecular relaxivity of LanND-Gd ($>52\,mM^{-1}\,s^{-1}$) exceeds that of Magnevist ($<5\,mM^{-1}\,s^{-1}$) by more than tenfold. Additionally, the protein-based contrast agents also displayed increased $r_2$ values (Supplementary Fig. 9). Notably, both $r_1$ and $r_2$ exhibited an inverse relationship with magnetic field strength, in contrast to the weak dependence of Magnevist. This observation, together with the improved relaxivity performance, is consistent with a previously reported MRI contrast agent derived from a $Ca^{2+}$-binding protein, which also coordinates approximately two water molecules[24,26]. The unique properties of LanND-Gd make it highly suitable for MRI applications at clinically relevant field strengths, which typically range from 1.5 T to 3 T.

### High-resolution brain imaging enabled by LanND-Gd
We next measured the structural size of LanND to be approximately $3.3 \times 3.7 \times 3.0\,nm^3$, which is smaller than that of most $Gd^{3+}$-based inorganic nanoparticles used in MRI (Fig. 2a)[11,27]. Hydrodynamic measurements indicate that both ion-free proteins and $Gd^{3+}$-bound proteins are smaller than 5 nm, which theoretically falls within the range of thorough excretion from the body (Fig. 2b)[28,29]. This characteristic reduces the risk of undesired retention and accumulation, thereby enhancing the contrast agent's safety and translational potential.

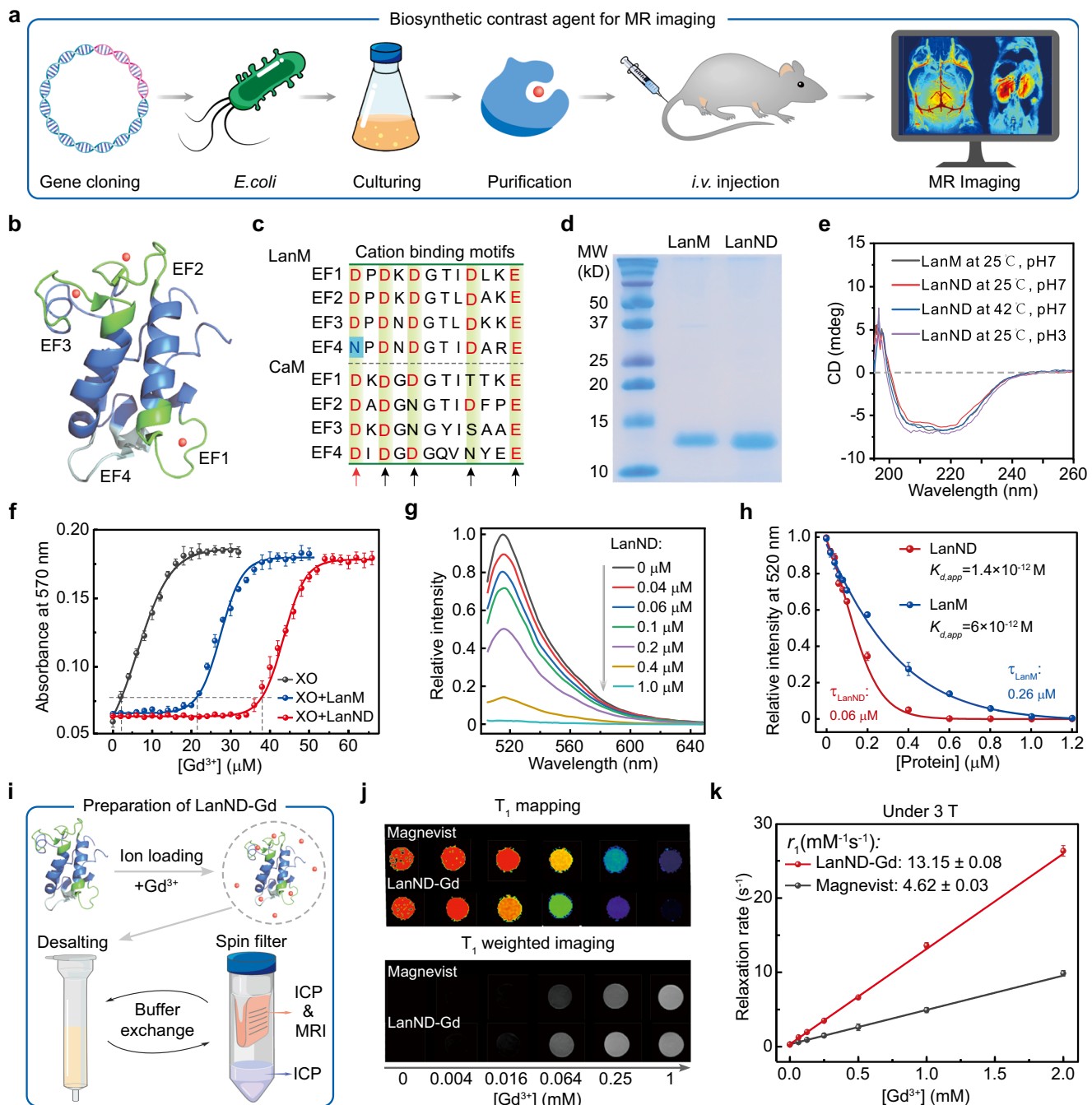

**Fig. 1 | Genetic optimization of lanmodulin-derived MRI contrast agents.**
**a** Schematic process of creating a protein-based MRI contrast agent for brain vasculature and kidney imaging. Lanmodulin was modified to improve protein yield by truncating the signaling peptide and to enhance metal loading capacity through single-point mutation. The modified gene was expressed in bacteria for purification. The proteins were loaded with $Gd^{3+}$ ions to create the contrast agent, which was administered intravenously for MRI examinations. **b** Structural analysis of wild-type LanM (PDB: 6MI5). A LanM molecule consists of three high-affinity lanthanide-binding sites (EF1, EF2, and EF3) and one pseudo-binding site (EF4) with lower lanthanide binding affinity. **c** Sequence alignment of the EF-hand motifs in calmodulin (CaM) and lanmodulin (LanM). The single-point mutation in LanM is indicated by cyan shading. **d** SDS-PAGE analysis of purified LanM and LanM_N108D (LanND). Experiments were repeated three times, yielding similar results. **e** Circular dichroism (CD) spectra of LanM and LanND under varying temperature and pH settings. **f** Xylenol Orange (XO) titration with $Gd^{3+}$ in the presence or absence of proteins. The concentrations of XO, LanM, and LanM-ND were set at 10 μM for all assays. Saturation of the tight binding sites for $Gd^{3+}$ was indicated by the absorbance at 570 nm reaching 10% of the total increase. **g** Fluorescence emission spectra of Fluo-5N (1 μM) in the presence of equimolar $Gd^{3+}$ (1 μM) and varying LanND concentrations. **h** Determination of apparent $Gd^{3+}$ binding affinity ($K_{d,app}$) for LanND by plotting fluorescence intensity against protein concentration. **i** Illustration of LanND-Gd preparation process. LanND and $Gd^{3+}$ were mixed to form the LanND-Gd complex, followed by buffer exchange to remove excess $Gd^{3+}$. **j** Comparison of $T_1$ mapping and $T_1$-weighted imaging between Magnevist and LanND-Gd at 3 T. **k** Correlation between relaxation rates and $Gd^{3+}$ concentrations at 3 T. Relaxivities were determined by linear fittings. Experiments were performed in biological triplicate and data are presented as mean ± SEM in (**f**, **h**, and **k**). Source data are provided as a Source Data file.

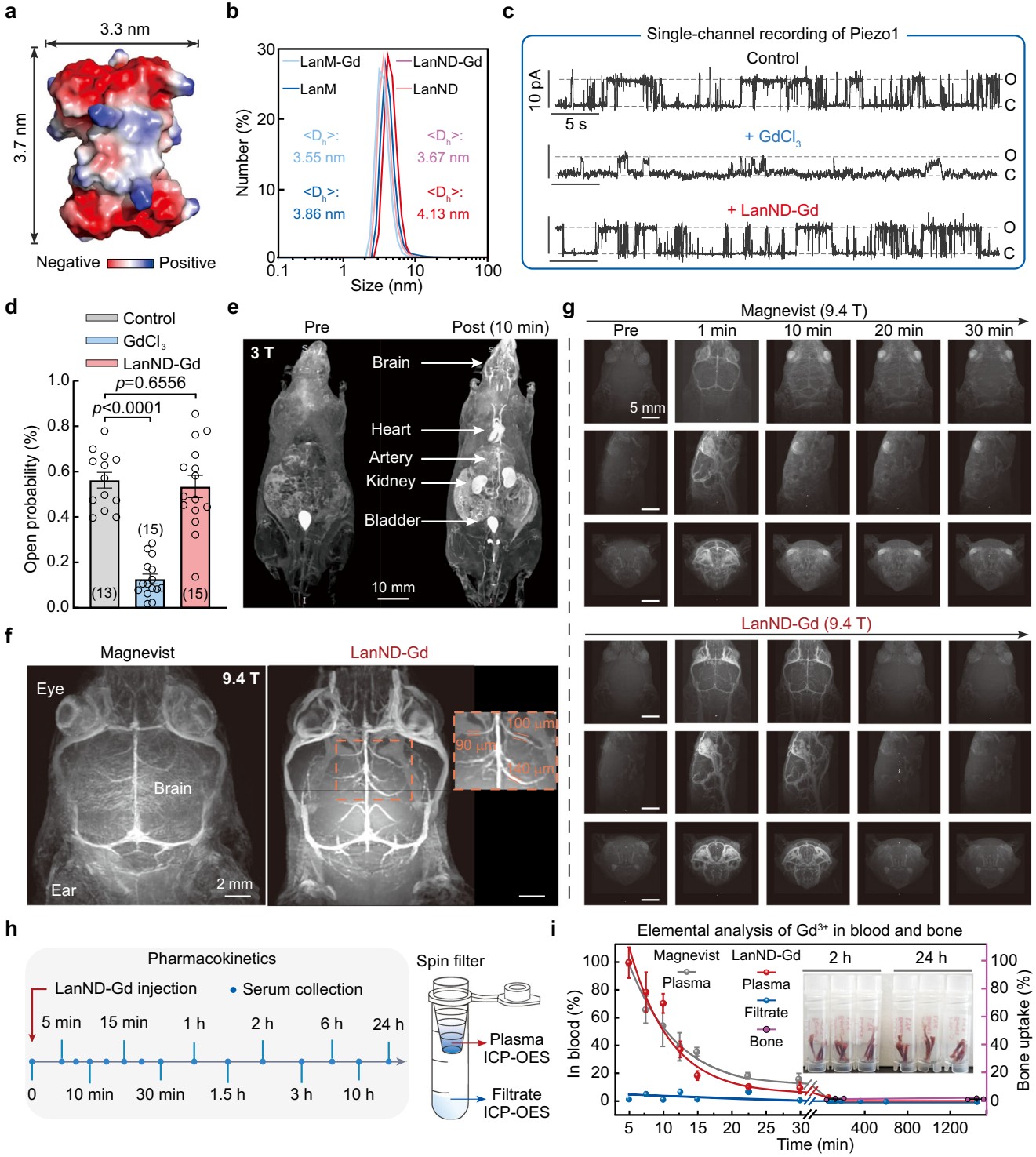

Prior to in vivo examination, it is crucial to ensure that $Gd^{3+}$ ions remain stably bound to the protein backbone and do not cause toxic effects under physiological conditions. We utilized the $Gd^{3+}$-sensitive mechanically gated Piezo1 ion channel to assess potential $Gd^{3+}$ ion leakage[30]. When we loaded the recording pipettes with 50 µM free $Gd^{3+}$ ions or LanND-Gd, we observed that free $Gd^{3+}$ ions significantly decreased the opening probability of Piezo1, whereas LanND-Gd had no discernible impact on channel activity, suggesting secure $Gd^{3+}$ binding within proteins (Fig. 2c, d). Moreover, when LanND-Gd or free $Gd^{3+}$ ions were co-incubated with Piezo1 expressing cells for over 24 h, $Ca^{2+}$ imaging results revealed that LanND-Gd

did not alter channel functions, while free $Gd^{3+}$ ions reduced channel responses to the agonist Yoda1 (Supplementary Fig. 10). Furthermore, we assessed the functionality of LanND in a complex environment, introducing interference factors such as $Ca^{2+}$, pH variations, temperature fluctuations and albumin to mimic in vivo conditions as closely as possible. The results of these competition titrations exhibited no differences from the control group (Supplementary Fig. 11). These observations indicated that $Gd^{3+}$ ions remain firmly bound to proteins and that LanND can sustain its normal function under complex environments, providing a solid molecular foundation for in vivo applications.

**Fig. 2 | LanND-Gd enables high-performance whole-body and brain MRI with a spatial resolution of 100 μm. a** Structural analysis of LanND. The structure of LanND was predicted using AlphaFold2. **b** Hydrodynamic size distribution of free and Gd-bound proteins. Experiments were performed in biological triplicate, yielding similar results. **c** Representative single-channel current traces of Piezo1 under various recording conditions. Cell-attached recordings were conducted and holding potential was maintained at +80 mV. "O" and "C" denote open and closed states, respectively. **d** Statistics of Piezo1 opening probabilities. Experimental numbers are indicated in parentheses, which is applied throughout the text unless otherwise specified. Data are presented as mean ± SEM. Unpaired student's $t$-test (two-tailed with criteria of significance: $^*p < 0.05$, $^{**}p < 0.01$, $^{**}p < 0.001$, $^{****}p < 0.0001$) were calculated. **e** $T_1$-weighted whole-body imaging of mice using LanND-Gd at 3 T MRI system. Images were acquired pre-injection and 10 min post-injection of LanND-Gd at a dose of 0.1 mmol $Gd^{3+}$ per kg. **f** Mouse heads scanned

after injection of Magnevist or LanND-Gd under a 9.4-T MRI scanner. **g** Comparison of $T_1$-weighted head images enhanced by Magnevist or LanND-Gd. Images are presented from longitudinal, side, and cross-sectional views. **h** Experimental timeline and sample processing for pharmacokinetic investigation of LanND-Gd. After injection of LanND-Gd, serum samples were collected from mice at multiple time points, and spin-filtered to separate LanND-Gd into retentate, with free ions in the filtrate. The retentate and filtrate were finally analyzed to quantify $Gd^{3+}$ concentrations. **i** Plots of relative $Gd^{3+}$ concentration in the retentate and filtrate, as well as in bones, over time. The normalized $Gd^{3+}$ concentration in the bloodstream for LanND-Gd was fitted using a monoexponential function ($\tau = 5.99 \pm 1.65$), with Magnevist used for parallel comparison ($\tau = 7.12 \pm 0.66$). Note that from 2 h to 24 h, ICP-OES indicated "not detected" as the $Gd^{3+}$ content fell below the limit of detection, so their values were set as "0". Data are shown as mean ± SEM, three mice were included for each time point. Source data are provided as a Source Data file.

After confirming the absence of cytotoxicity with the MTT assay (Supplementary Fig. 12), we intravenously administered LanND-Gd to mice at a dose of 0.1 mmol Gd per kg (equivalent to 25 μmol LanND-Gd per kg), following the clinically common dosing protocols for comparison with Magnevist. Subsequently, we acquired $T_1$-weighted 3D gradient-echo images of the entire body 10 min after injection using a clinical 3 T MRI system. MRI images exhibited marked enhancement of signal intensity in various organs, including blood vessels, heart, brain, and kidneys (Fig. 2e and Supplementary Movie 1). Notably, LanND-Gd exhibited effective contrast enhancement in cerebral vessels. We then evaluated the achievable resolution for brain vasculatures using LanND-Gd in a 9.4-T scanner. When the mouse head was scanned without contrast agents, only major parts, such as eyes, brain, and ears, could be visualized (Fig. 2g). After intravenous administration of LanND-Gd, intricate vascular structures as thin as approximately 100 μm in diameter became clearly visible (Fig. 2f and Supplementary Movie 2). Compared to the same imaging scans enhanced with Magnevist, LanND-Gd demonstrated generally superior imaging quality in terms of both signal-to-noise ratio and spatial resolution. Importantly, residual Magnevist accumulation persisted in the eyes during extended monitoring (Fig. 2g and Supplementary Figs. 13 and 14). In contrast, LanND-Gd was more thoroughly cleared from the brain without any signs of accumulation, suggesting a safer property associated with protein-based agents. It's worth mentioning that although the wild-type LanM-Gd exhibited comparable whole-body imaging results to LanND-Gd (Supplementary Fig. 15), there were experimental inconsistencies, persistent signals in the eyes, and higher mortality rates associated with LanM-Gd (Supplementary Fig. 16). These issues are likely due to the potentially hazardous release of $Gd^{3+}$ from the proteins, consistent with previous findings of radioactive $La^{3+}$ release from LanM-La[31]. To assess the pharmacokinetics and stability of LanND-Gd in the bloodstream, we collected serum at various time points after LanND-Gd injection. These samples were subjected to spin filtration with a cutoff size to retain LanND-Gd in the retentate while allowing free metal ions to enter the filtrate (Fig. 2h). When analyzing $Gd^{3+}$ content in the retentate and filtrate, as well as bones using ICP-OES, LanND-Gd circulated for around 1 h before the blood concentration dropped to a level below the detection limit (Fig. 2i). The circulation profile of LanND-Gd closely resembles that of Magnevist, indicating its normal pharmacokinetics as a molecular drug. Importantly, the $Gd^{3+}$ content in the filtrate and bones remained consistently low. This fact, combined with previous evidence of LanND's high stability in complex situations, suggests a strong likelihood that LanND-Gd circulates stably without notable breakdown or leakage of $Gd^{3+}$.

### Renal clearance of LanND-Gd
Following clearance from the bloodstream, we assessed the subsequent excretion pathway of LanND-Gd. Unlike conventional nanoparticle-based contrast agents, which are prone to endocytosis by cells or tissues during circulation in the bloodstream and subsequent

accumulation in organs, leading to concerns about potential side effects or toxicity[32,33], the compact size of LanND-Gd (<5 nm) falls within the range of complete urinary excretion from the body. On whole-body imaging, we observed prominent MRI signals in the kidneys and bladder, indicating effective filtration by the kidneys and subsequent excretion through the urinary tract (Supplementary Movie 1). Over the 0–75 min timeframe, MRI signals in the bladder, obtained from both cross-sectional and longitudinal abdominal imaging profiles, increased significantly and reached a plateau, confirming renal clearance through the urine (Fig. 3a, b). To further investigate the metabolic pathway of LanND-Gd, we performed ICP-OES for $Gd^{3+}$ in major organs. Following administration of the contrast agent and its circulation throughout the body, $Gd^{3+}$ ions were spread in multiple organs within 30 min. Subsequently, $Gd^{3+}$ was progressively distributed in the kidneys, while concentrations in other organs decreased significantly. After three days, Gd retention in the kidneys dropped to below 2%, and further decreased to less than 1% after approximately one week, suggesting the complete excretion of Gd from the body (Supplementary Fig. 17). To map the macroscopic biodistribution of LanND-Gd in animals, we genetically fused a fluorescent protein tag to LanND-Gd, resulting in GFP-LanND-Gd. This modification allowed real-time tracking of distribution throughout the body (Supplementary Fig. 18). After injection of GFP-LanND-Gd, the distribution of GFP fluorescence throughout the body was systematically monitored at various time intervals using the IVIS spectrum in vivo imaging system (Fig. 3c). Notably, the signal obviously accumulated in the bladder and then decreased, further supporting the safe urinary excretion route. The overall whole-body signal increased within 30 min and then gradually returned to baseline levels approximately 3 h after injection, suggesting that the contrast agent is excreted from the body within 3 h (Fig. 3d). Additionally, major organs, including the hearts, and lungs, livers, spleens and kidneys, were harvested for ex vivo organ imaging after injecting GFP-LanND-Gd (Fig. 3e). Similar to many intravenously administered drugs, the agent initially distributed temporarily in the lungs following the blood flow and then rapidly diminished within 30 min. In contrast, the signal distribution in kidneys remained relatively stable until 60 min post-injection, making it a favorable option for kidney function imaging (Fig. 3f). These findings obtained from $T_1$-weighted imaging, ICP-OES, and whole-body biodistribution analysis suggest that a significant portion of contrast agents is excreted via the renal route.

### Long-term monitoring of kidney dysfunction
Owing to the much larger molecular weight of LanND-Gd (~12 kD) compared with Magnevist (0.7 kD), its renal retention duration in the urinary system is expected to be prolonged, thereby increasing the effective imaging window. After the LanND-Gd injection, we monitored MRI signals in the kidneys to assess its retention time in organs, using Magnevist as a reference for comparison. Magnevist displayed brightened signals immediately after administration that lasted for

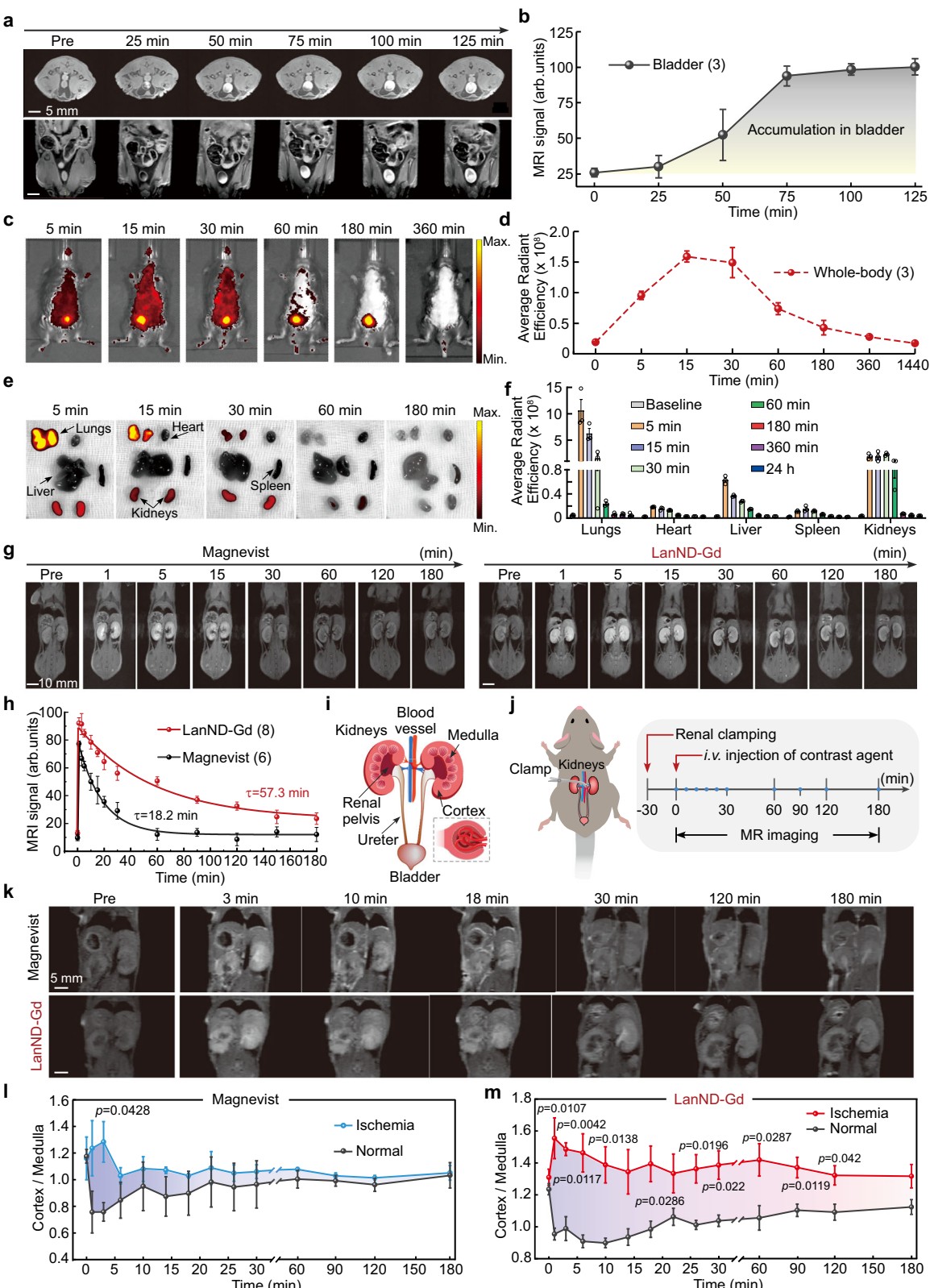

approximately 15 min. Thereafter, the signal diminished to a level hardly distinguishable from the pre-injection control (Fig. 3g). In contrast, kidneys administered with LanND-Gd displayed significant image enhancement as early as 30 min after administration, with discernible signals lasting up to 2 h (Fig. 3g). By analyzing the time-dependent decay of the average MRI signals from left and right

kidneys, we determined attenuation coefficients of Magnevist and LanND-Gd as 18.2 min and 57.3 min, respectively (Fig. 3h).

The prolonged renal retention time facilitates longer-term monitoring of kidney functions. Kidneys possess an intricate structure, consisting of an outer cortex, an inner medulla, and a renal pelvis (Fig. 3i). To investigate the efficacy of LanND-Gd in reporting kidney

**Fig. 3 | Renal clearable LanND-Gd offers the potential for long-term monitoring of kidney dysfunction. a** Urinary excretion of LanND-Gd verified by long-term bladder imaging. The top and bottom images display cross-sectional and vertical-sectional profiles, respectively. **b** Statistical analysis of the average MRI signals in the bladder against time. **c** Representative visualization of the whole-body distribution of GFP-tagged protein agents in mice using the IVIS imaging system. **d** Plot of the average whole-body signals over time, expressed in units of (p/s/cm²/sr)/(µW/cm²). **e** Representative distribution of GFP-tagged protein agents in major organs. **f** Statistical summary of average organ signals at each time point, expressed in units of (p/s/cm²/sr)/(µW/cm²). Each animal experiment was conducted with three independent mice. **g** Comparison of $T_1$-weighted imaging of kidneys with Magnevist or LanND-Gd as the contrast agent. Specific time points were selected to highlight differences in imaging outcomes. **h** Statistical analysis of temporal changes in MRI signals in the kidneys. The average signals of the left and right kidneys were applied, and the curves were obtained by exponential fittings. **i** Anatomical representation of kidney structure. **j** Experimental timeline for the establishment of the unilateral acute kidney injury (AKI) model and imaging. The left renal artery was clamped for 30 min to induce ischemia in the left kidney, while the right kidney was left unaffected as a parallel control. **k** MRI images of the AKI model obtained with Magnevist or LanND-Gd. **l** Statistical analysis of MRI signal changes in ischemic and normal kidneys contrasted with Magnevist. Experiments were conducted independently with three mice. **m** Statistical analysis of MRI signal changes in ischemic and normal kidneys contrasted with LanND-Gd. Each time point within 30 min included three independent mice, while four mice were involved for time points after 30 min. Unpaired student's t-test (two-tailed with criteria of significance: $^*p < 0.05$, $^{**}p < 0.01$) was calculated in (**l**, **m**). Data are presented as mean ± SEM in (**b**, **d**, **f**, **h**, **l**, and **m**). Source data are provided as a Source Data file.

dysfunction, we developed a unilateral ischemia-reperfusion injury mouse model[34,35]. The left kidney artery was completely clamped with a surgical hemostat for 30 min, followed by 30 min of reperfusion. Subsequently, LanND-Gd or Magnevist was intravenously administered to the animals, and imaging was conducted at various time intervals at a clinical 3-T magnetic field (Fig. 3j). Within 10 min, both left and right kidneys showed enhanced brightness. Both Magnevist and LanND-Gd showed clear differences between the left ischemic kidney and the right healthy kidney, with the medulla of the ischemic kidneys appearing darker compared to the uniformly bright signals in the right healthy kidney (Fig. 3k). As time progressed beyond the optimal time window for Magnevist, kidney images with Magnevist returned to original pre-injection levels, rendering kidney outlines indistinguishable, along with no discernible difference between injured and healthy kidneys. In contrast, with LanND-Gd, the discriminability of signals between the left damaged kidney and the right healthy kidney was preserved even 30 min after injection. Moreover, structural changes remained observable in images for several hours. When calculating the ratio of cortex to medulla for the ischemic and healthy kidneys, significant differences could be maintained with LanND-Gd for at least two hours, while Magnevist could only distinguish them within 10 min (Fig. 3l, m).

## Evaluation of biocompatibility, neural toxicity, and potential immune responses

We next investigated the biocompatibility of LanND-Gd in vivo. Following the injection of LanND-Gd into animals, they circulated throughout the body, reaching and permeating major internal organs. One week after imaging studies with LanND-Gd, the mice were sacrificed, and their organs, including the heart, lungs, liver, spleen, and kidneys, underwent histopathological analysis. Compared with the control mice, no abnormalities of cellular structures, such as cellular dilation, distortion, or collapse, were observed in the LanND-Gd-treated group (Fig. 4a). Moreover, there was no evidence of hemorrhage, inflammation, or necrosis, demonstrating the nontoxicity of LanND-Gd to these organs within the duration of the study. Since $Gd^{3+}$ has a similar ionic size to divalent endogenous cations, particularly $Ca^{2+}$, it competes with $Ca^{2+}$ and disrupts the normal function of $Ca^{2+}$-related signaling pathways, which can lead to toxicity in neural systems[36,37]. $Ca^{2+}$ signaling plays an essential role in neural systems. For instance, voltage-gated $Ca^{2+}$ channels in the cerebral cortex regulate neural excitability, synaptic transmission, and gene transcription[38,39]. To evaluate potential neural damage, we co-incubated LanND-Gd with isolated cortical neurons for 10 days. We traced neurite morphology, as indicated by overexpressed green fluorescent proteins in each neuron (Fig. 4b). Quantitative analysis revealed that the total neurite length was comparable to that of the control group treated with the same amount of phosphate-buffered saline (PBS, Fig. 4c). Additionally, comparisons using Sholl analysis showed that LanND-Gd had no

interference with the complexity of neural morphology, confirming its non-disruptive effects on neuronal maturation (Fig. 4d).

To determine whether LanND-Gd elicits immunogenicity, we monitored changes in body temperature following injection and found no signs of fever (Fig. 4e)[40]. These results indicate that the exogenous agents did not trigger acute inflammation. On the seventh day after the first administration, a sufficient period for immune memory generation, a second dose of LanND-Gd was injected to possibly elicit stronger immune responses, if any. Even with the second dose, there were no noticeable changes in body temperature. Daily food and water intake also showed no noticeable differences between the LanND-Gd injected group and the control group (Fig. 4f). The change in body weight also exhibited no obvious differences (Fig. 4g). Moreover, we investigated animals treated with two doses of LanND-Gd to examine a possible influence on behaviors due to immunogenicity or neural toxicity (Fig. 4h). Anxiety-like behaviors examined by open-box testing exhibited similar moving traces between the LanND-Gd injected group and the control group (Fig. 4i). Based on quantitative measurements, there were no observable differences in either the total retention time within center areas or the total moving distances between the two groups (Fig. 4j, k). Similar results were also obtained in depression-like behavior tests, including tail suspension, forced swimming, and elevated plus maze tests, further confirming that LanND-Gd did not induce any emotional perturbations within the two-week timeframe (Supplementary Fig. 19). Animals subjected to behavioral tests were sacrificed for brain slice analysis. Histological examination and Nissl's staining revealed no abnormities of neurons in various brain regions, including the hippocampus, amygdala, and hypothalamus (Fig. 4l). These observations suggest the non-toxic nature of LanND-Gd. Subsequently, we examined neural firing properties following behavior tests, which serve as a sensitive and primary indicator of neural activities. The results from the prefrontal cortex, a brain region closely related to emotion and memory, showed no discernible differences from the control group, including in neural firing frequency and the current threshold required to trigger action potentials (Fig. 4m–o).

LanND-Gd exhibited remarkable stability, remaining intact for over four months without aggregation, even in the presence of high salt concentrations. Importantly, its hydrodynamic size remained below the threshold for renal clearance (Supplementary Fig. 20)[28]. Furthermore, this protein-based contrast agent maintained consistent stability under various pH levels and temperatures, a finding supported by previous functional examinations using dye competition titrations (Supplementary Fig. 11). This exceptional stability can be largely attributed to the protein's ability to withstand harsh environmental conditions[18].

## Discussion

In summary, our research has yielded a high-performance MRI contrast agent derived from the recently discovered protein lanmodulin, which possesses a high affinity for lanthanides and increased water

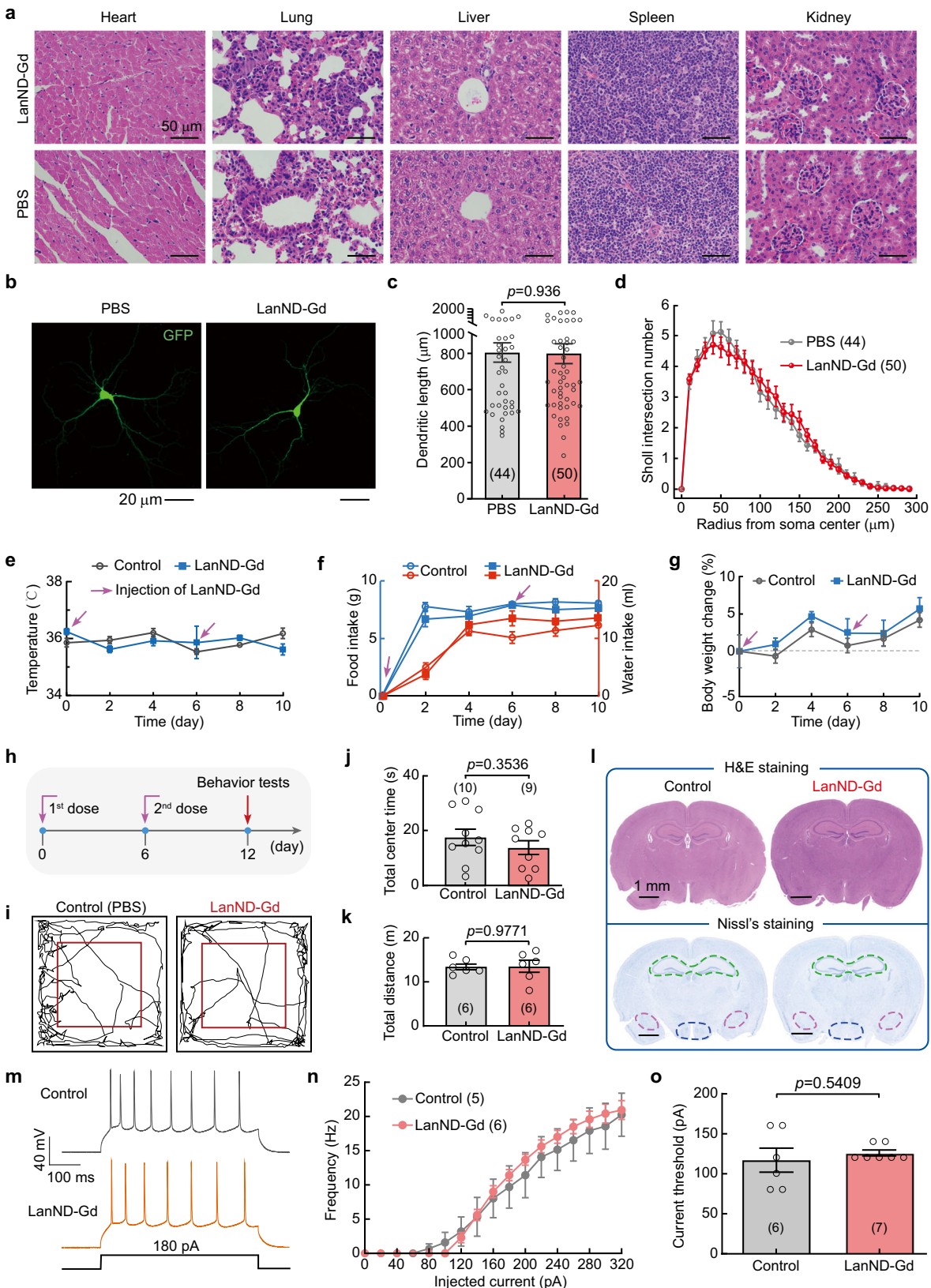

coordination number. Through a strategic single-point mutation, we have improved both $Gd^{3+}$-loading capacity and $Gd^{3+}$ binding affinity, resulting in a fully saturated four-binding site configuration. This modification minimizes the potential toxicity associated with the release of free ions and improves the safety profile of the contrast agent.

The compact size of our contrast agent, measuring below 5 nm, ensures its safe excretion through the urinary system and eliminates concerns about permanent retention in the body[41]. The enhanced relaxivity and prolonged renal retention time, when compared to commercially available contrast agents, have enabled high-resolution visualization of brain vessels and effective real-time monitoring of

**Fig. 4 | Assessment of biocompatibility and biosafety of protein contrast agents. a** Histological images of major organs obtained after intravenous injection of LanND-Gd. Experiments were repeated independently in three animal groups, yielding similar results. **b** Morphology analysis of cortical neurons incubated with either PBS or LanND-Gd. Neurons were transfected with GFP for neurite identification. **c** Quantification of total dendrite length. **d** Analysis of neuronal dendrite complexity. Sholl intersection numbers were quantified at different radii from the soma center. **e** Monitoring of body temperature during sequential injections of LanND-Gd ($n = 10$) or PBS ($n = 9$). **f** Dietary monitoring of animals during sequential injections of LanND-Gd ($n = 5$) or PBS ($n = 5$) to evaluate changes in eating behavior. **g** Body weight monitoring after intravenous injection of LanND-G ($n = 4$) or PBS ($n = 6$). **h** Experimental time points for neurobehavioral tests. **i** Analysis of open-field moving traces of animals subjected to sequential administrations of PBS or LanND-Gd. **j** Statistics of the duration of access to the central area during open field testing. **k** Total distance of exploration during open field testing. **l** Assessment of brain damage through histological (top) and Nissl's staining (bottom). Major brain regions, including the hippocampus (green dashed circle), amygdala (pink dashed circle), and hypothalamus (blue dashed circle) are highlighted. Experiments were biologically repeated in three animal groups, yielding similar results. **m** Representative traces of action potential firing in prefrontal cortex neurons from control or LanND-Gd injected group. **n** Statistical analysis of neural firing frequency induced by varying injection currents. **o** Summary of rheobase of the neurons. Data are presented as mean ± SEM in (**c–g**, **j**, **k**, and **n–o**). Unpaired Student's $t$-test (two-tailed with criteria of significance: $^*p < 0.05$) was calculated for (**c**, **j**, **k**, and **o**). Source data are provided as a Source Data file.

kidney function with a clear depiction of the cortex and medulla. The toxicity evaluation reveals no acute side effects associated with LanND-Gd. However, a more systematic investigation is certainly imperative to assess long-term safety issues and the potential Gd deposition in bones or skin before considering clinical translation. In addition to its exceptional utility in imaging fine vasculature and renal dysfunction, the lanmodulin-derived protein contrast agent also holds promise for the examination of other vital organs and various medical applications. It's noteworthy that currently used Gd-based MRI agents are unsuitable for patients with kidney diseases, as they may exacerbate kidney injuries. LanND-Gd may present a viable alternative, offering better biocompatibility and higher relaxivity. Moreover, this agent could substantially reduce the required dosage and further mitigate potential effects on kidney function. The high relaxivity, prolonged renal retention time, and biocompatibility of this contrast agent are properties that may benefit imaging in a broader medical context. The potential for further optimization, such as genetic modification with antibodies or pharmacokinetics-programmed peptides for lanmodulin, opens exciting possibilities for comprehensive morphological investigations in various diseases[42–44]. For instance, it can be employed to visualize liver, heart, vascular pathologies, or brain function with a level of detail and accuracy that surpasses conventional contrast agents. These advantages are particularly important in situations where high-resolution, dynamic, real-time monitoring of organ function with high-performance contrast agents is required.

## Methods

### Animals

All animal experiments, including MR scanning, pharmacokinetics studies, and behavioral tests, were conducted using male mice. Animal studies for MR imaging and behavior tests were conducted with the approval of the Animal Care and Use Committee of Shenzhen Institute of Advanced Technology, Chinese Academy of Science (animal protocol number: SIAT-IACUC-220921-YGS-SZH-A2188). In this study, we purchased male C57BL/6J mice aged 8–10 weeks from SPF Biotechnology Co., Ltd. Mice were kept in a controlled environment, with a consistent 12/12-h light-dark cycle and unrestricted access to standard food and water in cages for 5–6 mice. The ambient temperature was maintained at 23–25 °C. Random allocation was employed to divide mice into distinct groups.

For pharmacokinetics and biodistribution studies, we employed 8–10-week-old C57BL/6J male mice purchased from InVivos (Singapore). All mice were housed under special pathogen-free conditions with a 12/12-h light-dark cycle (lights on at 7 a.m., lights off at 7 p.m.) in the Comparative Medicine Animal Vivarium at the National University of Singapore. Standard chow diet and water were provided ad libitum. All studies were approved by the National University of Singapore Institutional Animal Care and Use Committee (IACUC; animal protocol number: R22-0407) and conformed to the guidelines on the care and use of animals for scientific purposes (NACLAR, Singapore, 2004) and the Guide for the Care and Use of Laboratory Animals published by the US National Institutes of Health (NIH Publication, 8th Edition, 2011).

### Materials and reagents

Chemical reagents, including isopropyl β-D-1-thiogalactopyranoside (IPTG), imidazole, phenylmethylsulfonyl fluoride (PMSF), and Coomassie Brilliant Blue-G250 were purchased from Golden Biotechnology Corporation. GdCl$_3$, Xylenol Orange, and 3-(4,5-dimethylthiazol-2-yl)-2,5-diphenyltetrazolium bromide (MTT) were purchased from Sigma-Aldrich. Ni-NTA agarose beads 6FF were obtained from Smart LifeSciences. All cell culture-related reagents were purchased from Life Technologies™ Singapore. All molecular biology reagents and competent *Escherichia coli* strain (*DH5α* and *BL21 (DE3)*) were obtained from New England BioLab Inc. Primers for gene cloning were purchased from Integrated DNA Technologies. Unless otherwise stated, all chemicals were used without further purification.

### Molecular biology

Wild-type lanmodulin (LanM, GenBank: CP001510.1) with a histidine tag (6 × His) at the C-terminus was synthesized by Tsingke Biotechnology (China) and inserted into the bacterial expression vector pET-24a with flanking NheI and BamHI. mPiezo1 was a generous gift from Dr. Jörg Grandl (Duke University Medical Center), and GCaMP6s was obtained from Addgene. We engineered truncations using standard molecular biology techniques (Forward primer: CTAGCTAG-CATGGCGCCAACT ACGACTACCAAAGTT; Reverse primer: CGCGGA TCCTTAGTGATGATGGTGATGATGGC CCTGAAA). The signaling sequence of lanmodulin was truncated to improve protein yield. Site-directed point mutation of lanmodulin_N108D (LanND) was cloned by overlap extension PCR, with the mutation designed into the overlapping primers (Forward primer: CTAGCTAGCATGGCGCC AACTAC-GACTACCAAAGTT; Overlapping reverse primer: TCAGGGTCAGCG GCCTTAAAC T G TGCCT; Overlapping forward primer: GCCGCTGACC CTGACAACGATGGCAC; Reverse primer: CGCGGATCCTTAGTGAT-GATGGTGATGATGGCCCTGAAA). sfGFP-LanND was constructed with overlap PCR to connect sfGFP to the N-terminal region of LanND (Forward primer: CTAGCTAGCATGCGTAAAGGCGAAGAGCTGTTCAC; Overlapping reverse primer: AGTCG TAGTTGGCGCACCTCCCCCT TTGTACAGTTCATCCAT; Overlapping forward primer: AACT GTA-CAAAGGGGGGAGGTGCGCCAACTACGACTACCAAAGT; Reverse primer: CGCGGAT CCTTAGTGATGATGGTGATGATGGCCCTGAAA). All segments subjected to PCR were verified by DNA sequencing.

### Protein purification

The proteins of lanmodulin-derived MRI contrast agents were purified with conventional purification methods. The plasmid containing the gene of interest, such as His-tagged wild-type LanM or mutant LanND, was transformed into the competent *Escherichia coli BL21 (DE3)* strain for expression. Then, the transformed bacteria were plated out on LB-agar plates containing 50 µg/mL kanamycin and grown at 37 °C. A single colony was picked for inoculation and grown for ~16 h at 37 °C

with a shaking rate of 200 rpm, after which the mixture was inoculated for large-scale culture. When the optical density at 600 nm ($OD_{600}$) reached 0.6, the culture temperature was set to 16 °C, and IPTG was added at a final concentration of 0.6 mM to induce the expression of recombinant proteins. After further overnight shaking incubation, bacterial cells were harvested by centrifugation at 4000 × $g$ and under 4 °C for 20 min.

Cells were lysed by sonication in a buffer containing 150 mM NaCl, 10 mM Tris·HCl, pH 7.5, and 100 μM PMSF. Thereafter, cell lysate was centrifuged for 30 min at a speed of 15,000 × $g$ at 4 °C, and the supernatant was collected to incubate with Ni-NTA agarose 6FF beads for 1 h at 4 °C. Notably, the Ni-NTA beads were pre-equilibrated in a buffer of 150 mM NaCl, 10 mM Tris·HCl, pH 7.5 (Buffer A). Then, the mixed solution was transferred to a gravity column and sequentially washed with 10-bed volumes of buffer A containing 10 mM or 20 mM imidazole. Subsequently, the proteins bound to the beads were eluted with 15-bed volumes of buffer A containing 200 mM imidazole. The eluted proteins were further concentrated using an Amicon Ultra 3-kD MWCO centrifugal filtration device and protein concentration was estimated by correction with an absorption coefficient obtained from ExPASy ProtParam (https://web.expasy.org/protparam/). Throughout the purification process, the protein-containing fractions were collected and finally analyzed by SDS-PAGE with protein staining by Coomassie Brilliant Blue. Based on purification results from multiple experimental batches, LanM demonstrates a protein yield of approximately 35 mg/L bacteria, while LanND shows a protein yield of about 40 mg/L bacteria.

For $Gd^{3+}$-loading into proteins, $GdCl_3$ was added to the proteins with a molar ratio of 6:1, then they were co-incubated for 30 min at 4 °C to ensure that metal ions were fully bound to the proteins. The unbound metals in the mixture were removed by exchanging buffer using Zeba Spin Desalting Columns (Thermos Fisher Scientific) and centrifugal filters (Millipore, MWCO: 3 kD). The metal contents in protein retentate and filtrated solution were quantified by ICP-OES.

## Dynamic light scattering and circular dichroism spectroscopy

Following the purification process, the proteins were dispersed in various buffers to reach a final concentration of 1 mg/mL. Subsequently, dispersed solutions were filtered using a 0.22 μm filter and the resulting samples were transferred to a cuvette for hydrodynamic size measurement. To improve accuracy, each measurement was conducted at least three times to ensure reliable and consistent results. Circular dichroism (CD) spectra analysis of proteins was conducted using a Jasco J-820 CD spectrometer and a 1-mm pathlength quartz CD cuvette (Jasco J/0556). Samples were scanned from 195 nm to 260 nm, under varying temperature or pH conditions with the same data collection parameters: 1 nm bandwidth, 0.5 nm data pitch, 50 nm/min scanning speed. All measurements were conducted for 50 μM proteins in a simulated physiological solution containing 135 mM NaCl, 5 mM KCl, and 2 mM $CaCl_2$.

## Xylenol orange (XO) competition experiments

Before competition experiments, buffer A for LanM and LanND was changed to a buffer containing 20 mM MOPS, 100 mM NaCl, and pH 6.0 (Buffer B). After quantifying the concentration of LanM and LanND, they were diluted to the same concentration of 100 μM by Buffer B. $GdCl_3$ and XO were also prepared to 100 μM with Buffer B. Three groups were applied for competition tests, one for control and the other two for LanM and LanND, respectively. Then, 10 μL of XO was first added to 96-well plates, after which 10 μL of buffer B, LanM, or LanND were added to corresponding wells to mix with XO. Thereafter, $GdCl_3$ was titrated into the wells at a final concentration of 0–70 μM with an alternating step of 2 μM. Different volumes of Buffer B were ultimately added to ensure that the final volume of each well was 100 μL. After all components were fully mixed, the plates were

transferred to read the absorption value at 570 nm using a microplate absorbance reader (Sunrise, Tecan). The absorbance was plotted against the metal ion concentration and fitted with the Boltzmann fitting. The metal concentration at the point of 10% saturation of XO was taken as an estimate of the point at which the tight binding sites within the proteins were saturated.

## Fluo-5N competition titration

The binding affinity of LanM or LanND to $Gd^{3+}$ was assessed through a competition titration with the $Gd^{3+}$-sensitive dye Fluo-5N (Invitrogen) as an indicator. First, the dissociation constant of Fluo-5N to $Gd^{3+}$ was determined with a Gd-NTA buffer system consisting of 20 mM HEPES, 100 mM NaCl, and 10 μM NTA at pH 7.5. The concentration of Fluo-5N was kept constant at 1 μM and $GdCl_3$ at concentrations from 0 to 100 μM was titrated into the system, resulting in a free $Gd^{3+}$ concentration from $10^{-13}$ to $10^{-8}$ M. The emission signal of Fluo-5N at 520 nm (excited at 488 nm) was measured and the $Kd1$ value of Fluo-5N for $Gd^{3+}$ was calculated using the Hill equation, resulting in a $Kd1$ value of $2.3 \times 10^{-11}$ M. Subsequently, the competition titration was performed by titrating various concentrations of LanM or LanND (from 0 μM to 1 μM) into the system containing 20 mM HEPES, 100 mM NaCl, 1 μM Fluo-5N, and 1 μM $GdCl_3$ at pH 7.5. The Fluo-5N emission spectrum from 505 nm to 650 nm (excited at 488 nm) upon the addition of LanM or LanND was monitored. The proteins can compete $Gd^{3+}$ out of Fluo-5N, leading to a decreased fluorescence signal when they are added. Finally, the $Kd$ value of LanM or LanND to $Gd^{3+}$ was calculated using Eq. (1):

$$Kd, app = \frac{Kapp \times Kd1}{Kd1 + [Fluo - 5N]} \qquad (1)$$

where $Kd1$ is the dissociation constant of Fluo-5N to $Gd^{3+}$, [Fluo-5N] is the concentration of Fluo-5N, $Kapp$ is the apparent dissociation constant of LanM or LanND to $Gd^{3+}$ obtained from the competition titration.

To investigate potential interference from various factors such as $Ca^{2+}$, Bovine Serum Albumin (BSA), pH changes, or temperature alterations, similar titration procedures were carried out, with the exception of adding appropriate amounts of these factors.

## Cell culture and transient transfection

The HEK293 cell line (ATCC) was exploited for electrophysiological recordings. Cells were cultured in DMEM, supplemented with 10% fetal bovine serum (FBS), 50 units/mL penicillin, 50 μg/mL streptomycin, and 5 μg/mL plasmocin prophylactically to prevent mycoplasma contamination. Cells were maintained in a water-saturated incubator with 5% $CO_2$ at 37 °C.

HEK293 cells were split and seeded in 60-mm dishes (at a density of $1 \times 10^4$ cell/cm²) containing coverslips to prepare for transfection 24 h–36 h before experiments. Transient transfection was carried out following the conventional calcium phosphate procedure. As for single-channel recordings or $Ca^{2+}$ imaging, 3 μg Piezo1 plasmids and 2 μg YFP or GCaMP6s were applied. Cells were washed with PBS 6 h after transfection and maintained in a culture medium of supplemented DMEM and then incubated for around 24 h before electrophysiology or imaging.

## Electrophysiology of cell-attached single-channel recordings

Borosilicate glass capillaries (Sutter Instruments, USA) were pulled with a programmable puller (P-1000, Sutter Instruments, USA) and heat-polished by a microforge (MF-900, Narishige, Japan), giving the electrode a resistance of 6–8 MΩ. The extracellular bath solution contained 140 mM NaCl, 10 mM HEPES, and 2 mM $CaCl_2$, with osmolarity adjusted to 290 mOsm/l by glucose and pH adjusted to 7.5 with NaOH. Bath solutions were controlled using Valve Commander ALA-

VM4 (ALA Scientific Instruments). The pipette solution for cell-attached recordings was obtained by adding 10 µM Yoda1 into the bath solution, with $GdCl_3$ or LanND-Gd further added for different testing groups. All recordings were conducted at room temperature with the Axon200B amplifier and the pCLAMP system (Molecular Devices). Electrical signals were amplified with the Axon 200B amplifier, low-pass filtered at 2 kHz, and digitized at 10 kHz.

For action potential recordings, the process involved preparing brain sections from deeply anesthetized mice using isoflurane. Subsequently, the brains were quickly decapitated and carefully removed from the skull. The brain was then fixed to a bed plate attached to an AGAR block of a vibrating microtome (VT1200s, Leica), and kept in ice-cold artificial cerebrospinal fluid (ACSF) containing (in mM): 2.4 $CaCl_2$, 3 KCl, 129 NaCl, 20 $NaHCO_3$, 1.3 $MgSO_4$, 1.2 $KH_2PO_4$, 3 HEPES, and 10 glucose. Coronal slices (300 µM) were cut at a controlled rate of 0.18 mm s$^{-1}$ and then incubated at a constant temperature (33 °C) for 30–60 min. The tissue sections were transferred to the recording room for electrophysiological recordings, with ACSF perfused at a specific rate of 2.5–3 mL/min under constant temperature (25 °C), maintained by an online solution heater (TC-344B, Warner Instrument). In ACSF, pH was set at 7.3–7.4, and osmolarity was adjusted to 300–305 mOsm/l. During section preparation and electrophysiological recording, ACSF was oxygenated continuously with 95% $O_2$ and 5% $CO_2$. Whole-cell patch-clamp recordings of prefrontal cortex neurons were performed with a ×40 water-immersion lens (BX51WI, Olympus) and an infrared-sensitive charge-coupled device (CCD) camera. The intracellular solution consisted of (in mM) 10 HEPES, 5 KCl, 130 K-gluconate, 0.6 EGTA, 2 $MgCl_2$, 2 Mg-ATP, and 0.3 Na-GTP, with osmolarity adjusted to 285–290 mOsm/l and pH to 7.2. Electrical signals were amplified with a MultiClamp 700B amplifier, low-pass filtered at 2.8 kHz, and digitized at 10 kHz.

### Fluorescence $Ca^{2+}$ imaging with GCaMP
After transfection of Piezo1 and GCaMP6s into HEK293 cells, 100 µM $GdCl_3$ or LanND-Gd was added to the culture medium and co-incubated with cells for 24 h. Then, the cells were transferred to a chamber on the inverted fluorescence microscope (Ti-U, Nikon, Japan), and the bath solution was the same as that used for single-channel electrophysiology. Epi-fluorescent images were collected with an EMCCD camera (Andor Technology). The light excitation source for GCaMP6s was a mercury lamp filtered at the appropriate wavelengths through optical filters and then passed through the dichroic mirror and emission filters. Parameter settings and image measurements were controlled by iQ3 software (Andor Technology). Fluorescence intensity was subtracted from the corresponding background, and the index $F/F_0$ was calculated to serve as an evaluation indicator. Of note, $F_0$ is the baseline fluorescence.

### Culturing of cortical neurons and transient transfection
Newborn ICR mice were used to dissect cortical neurons to evaluate neural toxicity. Cortical cortex tissue was isolated and digested with 0.25% trypsin for 15 min at 37 °C. Afterward, the enzymatic reaction was terminated by adding DMEM supplemented with 10% FBS. The resulting cell suspension was filtered and centrifuged at 1000 × g for 5 min. The cell pellet was resuspended in DMEM supplemented with 10% FBS and plated on poly-D-lysine-coated 35 mm confocal dishes at a density of $1 \times 10^4$ cells/cm$^2$. After 4 h, neurons were maintained in a Neurobasal medium supplemented with 2% B27 and 1% GlutaMAX (growth medium) at 37 °C and 5% $CO_2$ in an incubator.

After neurons had been cultured for 5 days, 2 µg of plasmids encoding GFP were transiently transfected into neurons with Lipofectamine 3000 (Invitrogen) according to the user manual. Opti-MEM containing the plasmids and Lipofectamine 3000 was added to the confocal dishes for transfection. After 2 h, neurons were maintained in a Neurobasal medium supplemented with 2% B27 and 1% glutaMAX-I.

One day after transfection, 200 µM LanND-Gd was added to the confocal dishes, and the same amount of PBS was added to the control dishes. After 10 days of co-incubation, confocal images were acquired for the analysis of neural morphology.

### Confocal imaging and data analysis
Confocal dishes containing neurons were imaged with a ZEISS laser scanning confocal microscope (LSM 800) (Carl Zeiss) and analyzed with ZEN 3.0 software. Neurite morphology was analyzed by measuring the total length and Sholl intersections with Imaris 9.0.1 (Bitplane). Only non-overlapping neurons were selected for images and analysis.

### MTT to measure cell viability
Cells were seeded into a 96-well plate at the density of $1.5 \times 10^4$ cells/well and cultured in the incubator for 48 h to allow the cells to settle to the bottom. After carefully removing the culture medium from the wells, MTT solution (5 mg/mL) was added to each well. Then the plate was incubated for 4 h at 37 °C, allowing MTT to be metabolized by viable cells into formazan crystals. Afterward, the solubilization solution of dimethyl sulfoxide was added to each well to dissolve the formazan crystals and liberate the dye. After further incubation for 20 min at room temperature, the plate was placed in the microplate reader to measure the absorbance of the dissolved dye at 570 nm.

### Establishment of the acute kidney injury model
The purchase male mice were maintained fasting for 12 h before surgery. After that, intraperitoneal anesthesia was induced with 0.2 mL of Avertin. Subsequently, the mice in the experimental group were fixed in the supine position on the operating table. After skin preparation and sterilization of the area, a longitudinal incision of approximately 3 cm was made near the 12th rib, 1 cm to the left of the midline. With the incision, the dorsal fascia was exposed. The fascia on both sides of the vertebral muscles was cut open. The perirenal fat tissue was bluntly dissected to fully expose the bilateral renal pedicles. The left renal pedicle was clamped using a noninvasive arterial clamp to induce 60 min of ischemia. The color of the kidney gradually changed from bright red to dark purple, indicating the successful establishment of the injury model. After surgery, the incision was closed with sutures. In the sham group, the renal pedicles were only bluntly dissected without clamping them. Postoperatively, all mice had free access to water and food. Of note, the criteria to confirm the accuracy of our modeling were as follows: when we clamped one side of the renal pedicle, a change in the kidney's color from reddish-brown to dark purple within 5 min indicated successful obstruction of blood flow. After releasing the microvascular clamp 60 min later, if the kidney changed from dark purple to reddish-brown within 5 min, this was considered successful reperfusion.

### MRI scanning
As for $T_1$-weighted determination of relaxation time, scan parameters for T1WI and T2WI under 3 T (uMR 790, United Imaging Healthcare, China) were as follows: radiofrequency coil, human 48-channel receiver head coil; TR, 500 ms; TE, 20 ms; FOV, 150 mm × 250 mm; FA, 145°; matrix size, 512 × 308; bandwidth/pixel, 200 Hz/pixel; slice thickness, 3 mm. For animal $T_1$-weighted imaging, the 3 T preclinical MRI system with the following specifications: radiofrequency coil, special small animal coil (mouse coil); TR, 600 ms; TE, 12.16 ms; FOV, 60 mm × 30 mm; FA, 145°; voxel size, 0.47 mm × 0.23 mm × 0.6 mm; matrix size, 256 × 256; bandwidth/pixel, 260 Hz/pixel; slice thickness, 0.6 mm; number of averages, 8; number of slices, 10. Measurements at 7.0 T were performed using a 7.0-T scanner (Bruker). And the parameters were set: imaging frequency (IF), 300.42 MHz; matrix size, 216 × 320; FOV, 83 mm × 81.25 mm; slice thickness, 0.45 mm; TE, 2.42; TR, 4000; TI, 1000.

As for brain imaging, a 9.4-T scanner (uMR 9.4 T, United Imaging Healthcare, China) was employed with the following specifications: coil, dedicated mouse brain surface coil; scanning sequence, GR; TR = 10.39 ms; TE = 1.75 ms; FOV = 18 × 22 mm$^2$; NEX = 1; FA = 20°; voxel size = 0.1 × 0.1 × 0.1 mm$^3$; matrix size = 224 × 183; slice thickness = 0.1 mm; number of slices = 160; bandwidth/pixel = 195 Hz/pixel; over-sampling = 10%; echo train length = 1.

## Pharmacokinetics study

As for the pharmacokinetics of gadolinium in the bloodstream, male mice were randomly divided into five groups for the pharmacokinetics study of the protein agent LanND-Gd. Two-time points were assigned to each group for whole blood collection via submandibular bleeding to cover time ranging from 5 min, 7.5 min, 10 min, 12.5 min, 15 min, 22.5 min, 30 min, 60 min, 90 min, 120 min, 180 min, 360 min, and 600 min after intravenous administration of LanND-Gd (0.1 mmol Gd per kg). Similarly, Magnevist was administered at the same dosage as LanND-Gd, and blood samples were collected at multiple time points (5 min, 7.5 min, 10 min, 12.5 min, 15 min, 22.5 min, 30 min, 60 min, and 90 min) to serve as a parallel control. Additionally, mice injected with saline were used as controls. The collected whole blood samples were transferred into EDTA tubes, and spun at 2000 × $g$ for 10 min at 4 °C to collect plasma. For each sample, 30 μL of plasma was diluted with 170 μL Buffer A and then underwent spin filtration to separate LanND-Gd into retentate, and the collected retentate and filtrate were analyzed to quantify Gd$^{3+}$ concentrations.

As for pharmacokinetics of gadolinium in major organs, the experimental procedure involved the following steps: at different time points after intravenous injection of LanND-Gd (0.1 mmol Gd per kg), mice were sacrificed and perfused with PBS. Subsequently, the heart, liver, spleen, lungs, kidneys, leg bones, and brain tissues were collected, weighed, digested, and diluted to a final volume of 10 mL by PBS. Finally, the Gd contents in these solutions were quantified by a Perkin Elmer Avio 500 ICP-OES.

## Biodistribution study

GFP-tagged LanND-Gd (0.1 mmol Gd per kg) was injected intravenously into 8–10-weeks-old wild-type male mice. The distribution of the GFP fluorescent signal in the whole body was measured at 5 min, 15 min, 30 min, 60 min, 180 min, 360 min, and 1440 min post-injection with the IVIS Spectrum in vivo imaging system (Perkin Elmer, Waltham, Massachusetts, United States). Additionally, another group of wild-type male mice was also injected with the GFP-LanND-Gd, and the hearts, lungs, livers, spleens, and kidneys were harvested at 5 min, 15 min, 30 min, 60 min, 180 min, 360 min, and 1440 min post-injection for ex vivo organ imaging. The images were analyzed with Living Image version 4.7.4 (Perkin Elmer, Waltham, Massachusetts, United States).

## Open-field test

Before conducting experiments, the mice used in the study were given an approximately 2-h acclimatization in the learning laboratory to minimize potential stress factors. During the actual experiment, each mouse was placed individually in the center of an open field with dimensions of 40 cm × 40 cm × 40 cm. They were allowed to freely explore the area for a duration of 5 min. A behavior recorder was used to record their movement trajectories during this period. For subsequent analysis, the central area of the field, defined as 24 cm × 24 cm, was identified. The time it took the mice to enter this central area and their total travel distance were analyzed offline with Smart v3.0 software.

## Tail suspension test

Throughout the experiment, a technique was employed involving taping the mouse's tail approximately 1 cm away from the tip. The mouse was then suspended 50 cm above the ground for 5 min. The experiment was recorded from a side-view perspective to capture the entirety of the process. Specifically, the total duration of immobility of the mouse during the 5-min period was recorded.

## Forced swimming test

A transparent glass cylinder with a height of 30 cm and a diameter of 19 cm was utilized for the experiment. The cylinder was filled with 24 °C fresh water and the water level was set at 25 cm from the bottom. The mice were carefully placed inside the cylinder and filmed from a side-view perspective for 5 min. The purpose was to record their resting periods. In this context, depressive behavior was characterized by the mice floating on the water's surface and making minimal movements to ensure that their noses remained above the water.

## Body temperature measurements

To measure the temperature of the mice, a small animal thermometer (Calvin, FT3400) was used. The thermometer was inserted into the anus of the mice after they were gently held. Once inserted, the temperature reading was taken after the temperature stabilized for 5 s.

## Monitoring of food and water uptake

Food monitoring: each mouse receives a pre-weighed amount of food in its food container. The containers are securely attached to the cage to prevent spillage. The amount of food given is recorded, along with the weight of the container. Water monitoring: fresh water is provided to the mice in water bottles attached to the cage. The bottles are filled with a known volume of water, and spillage or leakage is minimized. The initial water volume is recorded, along with the weight of the water bottle.

## Tissue staining

After injection of the protein agents or sham PBS, major organs, including the liver, spleen, lung, heart, kidney, and brain were collected. To prepare the tissue sections, the organs were cut into 8 μM thick pieces. H&E staining was then performed. Notably, some brain slices also underwent Nissl's staining. Ultimately, histological sections were viewed under an optical microscope.

## Statistics and reproducibility

Imaging data were analyzed using ImageJ, GraphPad Prism 10.1.2, and Origin 2019 software. Electrophysiological data were collected using pClamp 11.2 and analyzed in Clampfit 11.2 (Molecular device). Sequence alignment was conducted using Vector NTI software (Thermo Fisher Scientific). The structure was predicted using Alpha-Fold2 and analyzed in PyMOL. Standard error of the mean (S.E.M.) and student's $t$-test (two-tailed with criteria of significance: $^*p < 0.05$; $^{**}p < 0.01$, and $^{***}p < 0.001$) were calculated when applicable. No statistical method was used to predetermine sample size, and no data adjustments were made from the analyses.

## Reporting summary

Further information on research design is available in the Nature Portfolio Reporting Summary linked to this article.

## Data availability

Protein sequence and structure data were retrieved from the Protein Data Bank: lanmodulin (6MI5 [https://doi.org/10.2210/pdb6MI5/pdb]). All the data used to generate figures in this paper are provided in the Supplementary Information and Source Data file. The full image dataset is available from the corresponding author upon request. Source data are provided with this paper.

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

## Acknowledgements

This work was supported by the National Research Foundation, Prime Minister's Office, Singapore, under the NRF Investigators Program (X.L.: Award No. NRF-NRFI052019-0003), NUS NANONASH Programme (X.L.: NUHSRO/2020/002/NanoNash/LOA; R143000B43114), National University Health System Seed Fund (J.W.: NUHSRO/2018/095/RO5 + 5/Seed-Nov/05), MOE Tier 1 (J.W.: NUHSRO/2021/113/T1/Seed-Sep/06), NUS Yong Loo Lin School of Medicine Nanomedicine

Translational Research Program (J.W.: NUHSRO/2021/034/TRP/09/ Nanomedicine), the Natural Science Foundation of China (C.C.: 92159304, 82372022, T2241002 and 12225511), Key Laboratory for Magnetic Resonance and Multimodality Imaging of Guangdong Province (Z.S.: 2023B1212060052), Guangdong Basic and Applied Basic Research Fund (Z.S.: grant no. 2023A1515010747), Science and Technology Key Project of Shenzhen (Z.S.: JCYJ20210324120011030). We thank Dr. Min Luo and Dr. Xin Xu from the National University of Singapore for technical assistance. S.Y.C. would like to thank the generous support from the ESR and Loo Geok Eng Foundation PhD scholarship.

## Author contributions

X.L. and Y.L. conceived and designed the project. X.L., H. Z., and Z.S. supervised the project and guided the collaboration. Y.L. performed molecular biology, protein purification, electrophysiology, and protein characterization. Y.L. and J.T. optimized protein expression. D.G., Z.L., and Z.Y. performed in vivo MRI imaging experiments. Y.H. and J.M. conducted neurobehavioral tests. S.Y.C., X.Q., and H.J.T. performed pharmacokinetics and biodistribution studies. C.C. and J.W. supervised behavioral experiments. Y.L., D.G., Y.H., and S.Y.C. performed data analysis. Y.L. and X.L. wrote and edited the manuscript. All authors participated in the discussion of the manuscript.

## Competing interests

X.L., Y.L., and Z.L. are co-inventors on a Singapore patent application related to the use of lanmodulin-derived contrast agents for magnetic resonance imaging (application no. 10202400550P), filed by the National University of Singapore. The other authors declare no competing interests.
