## [Transparent Peer Review file · Nature Communications]

Single-Point Mutated Lanmodulin as a High-Performance MRI Contrast Agent for Vascular and Kidney Imaging

Corresponding Author: Professor Xiaogang Liu

Version 0:

Reviewer comments:

Reviewer #1

(Remarks to the Author)

In this manuscript, the authors generate a N108D mutant (LanND) of the lanthanide-binding protein lanmodulin (or LanM) and use it as a Gd-based MRI contrast agent. LanM has four EF hands but three of them are much tighter than the fourth. These EF hands bind lanthanides with 2 coordinated water molecules, which suggests that the protein would perform well for MRI contrast. The authors make the N108D mutation to increase the affinity of EF hand 4, thereby hopefully increasing binding stoichiometry and increasing relaxivity. The resulting protein is reported to bind 4 Gd per protein and its MRI contrast properties are compared with Magnevist, a classic MRI contrast agent. In many ways, the protein seems to perform better than Magnevist in in vivo studies, with higher signal and faster clearance.

There are two important points that I believe the manuscript needs to address.

Major comments:

First, does the LanND protein behave any better in vivo than the wildtype LanM? There are some hints that, while LanND may bind Gd slightly better in vitro, its folding might be somewhat disrupted (the hydrodynamic radius of LanND in Figure 2b is significantly higher than a previous report for LanM). Not all of the in vivo experiments would need to be repeated with LanM, but I think the basic ones should be.

Second, the authors do not show direct evidence that the Gd remains stably bound to the protein, only that it does not end up in the small molecule fraction of the serum. This is a concern because recent work from the Boros group (<https://pubs.rsc.org/en/content/articlehtml/2023/cb/d3cb00020f>) has shown that some of the radioactive La³⁺ bound to LanM injected into mice remains bound after 2 hours, but some of it dissociates and gets taken up by bone. (That work should be cited by the authors and this work discussed in the context of those results.) LanM binds La and Gd similarly. Therefore, unless the performance of LanND is significantly better than that of LanM, I would expect that it too is losing some, but not all, of its Gd.

If it turns out that LanND's performance is similar to LanM's, this work is still very interesting. However, a significant amount of the work would need to be discussed differently.

Other comments:

Introduction: it is probably worth mentioning that it is not just LanM's high affinity and selectivity that make it worth investigating as a GBCA – it is that each of the metal binding sites has two coordinated water molecules ($q=2$).

Line 80: The authors say that the much lower affinity of EF hand 4 is a problem for in vivo applications. However, it is easy to not have EF hand 4 occupied because metal bound to it can be removed by gel filtration before even being put in vivo. However, I agree that a higher stoichiometry would be better than a lower one.

Figure 1a legend: are the authors implying from the text and figure that LanND gives a better protein yield than LanM? If so, please give the yields in the Methods.

Are the CD spectra in Figure 1e for the apo or Gd-bound proteins? These spectra look a bit different from previously published spectra for LanM. The features at 205 and 220 nm or so do not seem distinct as they do for LanM in the literature (e.g. ref 22).

It is a bit confusing that Figure 1h mentions apparent Kds in the micromolar range. I understand that this is in the context of a competition assay with Fluo-5N, but I think this could be clarified for readers who may be less familiar with the determination of Kds for tight binding molecules. How many binding sites on the protein did the authors assume for their Kd calculations?

Figure 1k: how much Gd was bound to LanND after filtration and ICP-OES in the protein used for this experiment?

In the conditions in Supplementary figure 4, the wild type LanM binds roughly 4 equivalents and the plot at left does not look very different from the LanND up to 1.5 mM protein concentration. On the plot at right, LanND binds roughly 5 equivalents Gd, but the protein would only be expected to have 4 binding sites. To me, this figure does not clearly demonstrate that LanND binds significantly more Gd than LanM under these conditions.

In Supplementary figure 6, the proteins with Gd added are cloudy, which seems not ideal. At what concentration and in what buffer was this experiment done?

MRI section, p. 4. Some relaxation data relevant to performance of the protein as a Gd MRI contrast agent has been reported for LanM (<https://pubs.acs.org/doi/10.1021/jacs.1c07103>). The prior work should be cited and the authors should discuss their results in the context of the WT protein. Also, because Magnevist only has $q=1$, I wonder whether comparison in the text to an agent with $q=2$ might be more fair? (No need to do experiments with another small molecule; I think discussion in the text would be sufficient.) I also think that a more extensive comparison in the text to the Yang group's protein-based MRI contrast agent (reference 27) would be appropriate.

Figure 2b and p. 5, line 132: a hydrodynamic diameter of 4.7 nm for LanND is quite high. In reference 23, the hydrodynamic diameter of LanM is reported as 3.5 nm, which is reasonable given the NMR and crystal structures. Could this mean that the LanND protein is not folded as fully as LanM?

Line 159: Can the authors clarify whether the same amount of Gd was injected in the case of the Magnevist studies as with the Gd-LanND?

Lines 166-168: these figures should be 2h and 2i, not 1h and 1i

Line 169 "demonstrating the stable circulation of LanND-Gd without detectable breakdown or leakage of Gd³⁺": While Gd is not detected in low molecular weight fraction of the serum, it does not necessarily mean that all the Gd still in circulation is still associated with LanND. See additional information in the major comment above.

Renal clearance section, p. 6: this work should be discussed in the context of the work by Boros, mentioned earlier. The kidney uptake is similar to what was observed for LanM injected with radioactive La, where bone deposition was also observed. Did the authors look for uptake of Gd by bone, which would be clear indication of Gd dissociation from the protein?

Reviewer #2

(Remarks to the Author)

This is a *magnus opus* describing the synthesis and extensive testing of a new customized protein, lanmodulin, that binds 4 gadolinium atoms. It's a smart and timely idea. The protein outperforms a clinically widely used single gadolinium chelate (Magnevist) in terms of relaxivity (efficacy of contrast enhancement), has a "longer" blood half-life, and better anatomical details on imaging. The study is very comprehensive. The inclusion of GFP to validate the MRI with optical imaging is also very nice.

I have a few major and minor comments that need to be addressed.

Major:

-There are some questions about blood clearance and retention, in fact I am puzzled by Fig. 2g, where the single chelate Magnevist appears to have a longer blood half life compared to Gd-lanmodulin, reversing all conclusions on an increased blood half-life of the latter. It is just said that Magnevist accumulates in the eye (I may be ignorant, but have never heard of it) but there are other tissues showing a prolonged contrast enhancement (longer than Gd-lanmodulin) in all 3 imaging planes. This cannot be ignored.

-Please calculate the blood half life from Fig 2i and provide this. Is curve fitting mono- or biexponential? Does it have a short clearance and a long clearance component or just one. More time points may be needed for a good fit but this is a very simple and straightforward experiment.

-How much Gd is eventually retained? This will be important long term. Collect all urine, measure Gd and subtract from the amount injected. Is 99.9% excreted at the end? Even a few % retainment may be of concern, eventually.

-All potential toxicity tests are performed very quickly after injection (1 week). Possible side effects will manifest themselves

much later, esp in the CNS. It is understood that long-term follow up studies are beyond the scope of this work, but there should be a clear limitation statement that the behavioral and toxicity tests may tell us not as much as the authors claim.

-Stability/dechelation/Kd max studies in plasma would also be valuable.

Minor:

-Line 119: add "molecular" between effective and relaxivity for clarity.

-Line 122: inverse relationship of r_1 and r_2 . This is not "Interestingly", much literature on T1 NMRD profiles exist since the mid 1980s. PMID: 11902904 describes a study on a Gd-protein (HSA) and the r_1/r_2 relationship on field dependence.

-Line 174: "nanoparticles circulate excessively". What is meant with "excessively"? This suggests extensively, or a long blood half life? But they have a very short half live before taken up by the RES, much shorter than Gd-lanmodulin. Please remove or rephrase.

-Line 184: "Accumulation" in kidneys. This suggests uptake. It merely passes through. "Distribution" may be a better term.

-Line 196: This protein is not a nanoparticle and the lung uptake is unexpected. Needs further thought. But even nanoparticles do not accumulate in lungs. Mostly liver, spleen, and lymph nodes.

-Line 201. The kidney itself is not the primary route of excretion. That is the bowel, where the term "excrements" has been coined from.

-Line 203 and 216: It is (dys)function, not (dys)functions.

-Figure 3e. Please provide abbreviations indicating organs. One has to sort of distill it from panel 3f.

Reviewer #3

(Remarks to the Author)

Thank you for asking me to review this manuscript in which a new MRI contrast agent is described, based on a protein (LanM) that is modified with a single point mutation and is then saturated with gadolinium. Results in terms of imaging show that there is a reduction in relaxation times, resulting in brighter images from different organs and better resolution of the vasculature as compared to standard gadolinium contrast (Magnevist). I am reviewing this from the point of view of a clinician; it will be important to ensure that there is sufficient review of the MRI physics aspects.

The main potential weakness relates to the renal elimination of the new contrast agent, and that all of the experiments were performed in animals with normal renal function (in particular toxicity studies). One of the main limitations of standard gadolinium (in particular in research settings) is its use in patients with impaired kidney function – in contrast the use of gadolinium in patients with normal kidney function is generally viewed as much lower risk. It is possible therefore that due to reduced clearance, the toxicity profile may be substantially different/worse in those with impaired renal function, and this new agent may end up having exactly the same issues as standard gadolinium.

In the introduction, it is not clear what the aim of the new contrast agent is – shortening relaxation times, visualisation of vasculature and evaluation of kidney function are all mentioned, but more precision about the aims of developing this specific contrast agent would be helpful.

There is a lack of justification of how and why LanM was chosen, and this should be included in the introduction.

It is possible that gadolinium dissociates from LanM after infusion, so it was good that ex vivo experiments were included that suggested that this was less likely.

Suppl Fig 13 – although the results are reported as non-significant, there are some concerning patterns seen in relation to potential toxicity. Somewhat offset by toxicity studies based on histology although these will not capture longer term or functional effects. This should be discussed and may need further studies to reassure on this point.

Version 1:

Reviewer comments:

Reviewer #2

(Remarks to the Author)

Thank you for making the many revisions. Coming back to my previous point, I am still confused about previous Fig 2g and new Suppl Fig 14. Outside of the eye, other tissues clearly have a longer residual (10,20, 30 min) diffuse contrast for Magnevist compared LanND-Gd, contradicting the said longer blood-half life of the latter. This applies to all 3 mice. Throughout the manuscript, it is stated at different places that LanND-Gd has a longer retention (which seems to have been interpreted here as having a longer blood half life). "Retention" normally refers to the residual cell/tissue amount after having been cleared from blood

This agrees with the very short half life of LanND-Gd that has now been calculated (thank you) from Fig 2i as being 6 minutes. This is much shorter than that of Magnevist (which has a biexponential fit with 12 and 96 min half-life values in

humans). Magnevist clearance data for mice are not given here (could do the same expt as Fig 2i). What is this paradox?

Reviewer #3

(Remarks to the Author)

Thank you - all of my points have been addressed

Reviewer #4

(Remarks to the Author)

As a substitute reviewer, I find that the manuscript presents extensive work on the modification of lanmodulin for use as an MRI contrast agent. The results are impressive, and the effects of a single point mutation are fascinating. The reviewers' questions were well thought out and demonstrated a deep engagement with the manuscript.

I would have appreciated more characterization of the mutation effects. For instance, the changes in the affinity of the EF-hand compared to LanM could have been assessed through mutations aimed at reducing the affinity of the highly affine sites 1 to 3, which would have simplified the data complexity. However, I acknowledge the breadth of work completed and recognize that the focus was more on the application in the organism.

I believe the review has helped improve the quality of the manuscript. The authors have addressed all comments appropriately, responding with relevant experiments and clearer textual representations. I have no further objections to the publication.

Version 2:

Reviewer comments:

Reviewer #2

(Remarks to the Author)

The 2nd revision of the manuscript seems to have addressed my comments.

REVIEWER COMMENTS (Manuscript # NCOMMS-24-06716-T)

Reviewer #1 (Remarks to the Author):

In this manuscript, the authors generate a N108D mutant (LanND) of the lanthanide-binding protein lanmodulin (or LanM) and use it as a Gd-based MRI contrast agent. LanM has four EF hands but three of them are much tighter than the fourth. These EF hands bind lanthanides with 2 coordinated water molecules, which suggests that the protein would perform well for MRI contrast. The authors make the N108D mutation to increase the affinity of EF hand 4, thereby hopefully increasing binding stoichiometry and increasing relaxivity. The resulting protein is reported to bind 4 Gd per protein and its MRI contrast properties are compared with Magnevist, a classic MRI contrast agent. In many ways, the protein seems to perform better than Magnevist in in vivo studies, with higher signal and faster clearance.

Response: We sincerely appreciate the reviewer for summarizing our work and highlighting lanmodulin's capability to coordinate two water molecules. We have revised our manuscript to emphasize this distinctive property throughout the text.

There are two important points that I believe the manuscript needs to address.

Major comments:

First, does the LanND protein behave any better in vivo than the wildtype LanM? There are some hints that, while LanND may bind Gd slightly better in vitro, its folding might be somewhat disrupted (the hydrodynamic radius of LanND in Figure 2b is significantly higher than a previous report for LanM). Not all of the in vivo experiments would need to be repeated with LanM, but I think the basic ones should be.

Response: We appreciate the valuable suggestions given by the reviewer. In the revised manuscript, we have evaluated the performance of LanM-Gd in whole-body imaging to compare it with

LanND-Gd under a 3.0T scanner. Our results indicate that LanND-Gd exhibits slightly brighter contrast than LanM-Gd (**Supplementary Fig. 15**), which may be attributed to the higher molecular relaxivity of LanND-Gd ($\sim 52 \text{ mM}^{-1}\text{S}^{-1}$ for 4 binding sites) compared to LanM-Gd ($\sim 39 \text{ mM}^{-1}\text{S}^{-1}$ for 3 binding sites) (**Supplementary Fig. 7**). When compared with Magnevist, both LanM-Gd and LanND-Gd show brighter images, and LanM-Gd is also excreted out of the body *via* urinary route.

However, the difference between LanM-Gd and LanND-Gd is not very pronounced in whole-body scanning tests. Then we proceeded to compare the imaging performance of LanM-Gd and LanND-Gd under 9.4T scanning conditions, which are expected to provide more detailed information (**Figure 2g and Supplementary Fig. 16**). Unlike LanND-Gd and Magnevist, LanM-Gd exhibited inconsistency when serving as the contrast agent. We conducted brain imaging on four mice, two of which died during the imaging process, and the remaining two showed inconsistent results in effective time windows. We attribute this inconsistency to the unstable binding of Gd^{3+} in EF4, which may result higher possibility of Gd^{3+} release and cause variability between different experimental batches, and sometimes even lead to increased mortality.

Furthermore, we observed brighter signal retention in the eyes compared to LanND-Gd (**Figure 2g and Supplementary Fig. 14 and Supplementary Fig. 16**). These observations collectively suggest that LanM-Gd is less stable than LanND-Gd, highlighting the significance of single-point mutation in developing the new MRI contrast agent.

And for the hydrodynamic sizes, we remeasured them using a refreshed buffer system as previous report, with LanM and LanM-Gd as the controls. The results demonstrated comparable size distributions ($<4 \text{ nm}$) for LanM, LanM-Gd, LanND, and LanND-Gd (**Figure 2b**). Updated data and discussions related to above issues have been included in the revised text.

Supplementary Fig.7 | Characterization of the relaxivity performance of LanM-Gd.

Supplementary Fig.15 | Whole-body imaging contrasted with different contrast agents.

Supplementary Fig.16 | Brain imaging enhanced with LanM-Gd under 9.4T.

Figure. 2b, Hydrodynamic size distribution of free and Gd-bound proteins

Second, the authors do not show direct evidence that the Gd remains stably bound to the protein, only that it does not end up in the small molecule fraction of the serum. This is a concern because recent work from the Boros group (<https://pubs.rsc.org/en/content/articlehtml/2023/cb/d3cb00020f>) has shown that some of the radioactive La^{3+} bound to LanM injected into mice remains bound after 2 hours, but some of it dissociates and gets taken up by bone. (That work should be cited by the authors and this work discussed in the context of those results.) LanM binds La and Gd similarly. Therefore, unless the performance of LanND is significantly better than that of LanM, I would expect that it too is losing some, but not all, of its Gd.

Response: Yes, the reviewer's comments are highly constructive. We examined MRI for the bones and didn't detect any noticeable bone MRI signals. Then we evaluated Gd^{3+} uptake by the bones through element analysis following 2-hour and 24-hour injection of LanND-Gd. We extracted and digested the leg bones for ICP-OES after weighing, and found minimal Gd^{3+} accumulation in the bones (less than $0.5 \mu\text{g Gd/g bone}$, **Figure 2i**). It's known that clinically used Magnevist, with high Gd^{3+} binding affinity and fast clearance rate, also exhibits Gd^{3+} deposition in bones (up to $1.77 \mu\text{g Gd/g bone}$, Doi: 10.1038/s41598-020-63325-9). It's still challenging to determine the precise nature of this Gd^{3+} deposition to be free or agent-bound state. Although Gd^{3+} deposition needs further systematic investigation in the future, the Gd^{3+} uptake by bones have been largely reduced compared with Magnevist, indicating a potential safer profile of LanND-Gd than that of Magnevist.

We concur that this is an important concern, additional discussion and citation of the reference paper in this regard have been updated at corresponding positions of the revised manuscript.

Figure 2i, Plots of relative Gd³⁺ concentration in the retentate and filtrate, as well as in bones, over time.

If it turns out that LanND's performance is similar to LanM's, this work is still very interesting. However, a significant amount of the work would need to be discussed differently.

Response: We appreciate the positive comments. LanM and LanND clearly demonstrate distinct performance in ion binding affinity and stoichiometry. While they could not show pronounced differences in whole-body imaging under 3T, the closer examination of brains imaging under 9.4T accentuates the differences, where LanM-Gd is less stable in imaging outcomes and may cause higher mortality rate compared to LanND-Gd (as evidenced by above response and corresponding explanations in the revised manuscript).

Other comments:

Introduction: it is probably worth mentioning that it is not just LanM's high affinity and selectivity that make it worth investigating as a GBCA – it is that each of the metal binding sites has two coordinated water molecules (q=2).

Response: We have revised our introduction section and incorporated a sentence to emphasize the protein's unique property of coordinating two water molecules. The revised sentence reads as follows "The discovery of a high-affinity lanthanide-binding protein, lanmodulin (LanM), has inspired us to develop a high-performance MRI contrast agent owing to its ability to coordinate two water molecules. This unique property holds the potential to enhance the relaxivity outcome compared to commonly used clinical agents, which typically only coordinate one water molecule. Hence, we have engineered a LanM-derived protein MRI contrast agent tailored for high-resolution vascular imaging and effective monitoring of kidney dysfunction (Fig. 1a)."

Line 80: The authors say that the much lower affinity of EF hand 4 is a problem for in vivo applications. However, it is easy to not have EF hand 4 occupied because metal bound to it can be removed by gel filtration before even being put in vivo. However, I agree that a higher stoichiometry would be better than a lower one.

Response: Yes, it's a reasonable option. However, very rigorous experimental procedures are needed to ensure that gel filtration fully removes the weakly bound Gd^{3+} in EF4. As shown in **Supplementary Fig. 4**, the ratio between Gd^{3+} and LanM exceeds 3 (around 3.84), indicating that desalting column and spin filters cannot easily remove all weakly bound Gd^{3+} in LanM. Therefore, we optimally chose point mutation to saturate the binding sites, so as to minimize the risk of potential metal leakage. Additionally, we agree with the reviewer that higher stoichiometry can enhance molecular relaxivity to improve image outcomes, including both brightness and resolution.

Figure 1a legend: are the authors implying from the text and figure that LanND gives a better protein yield than LanM? If so, please give the yields in the Methods.

Response: We apologize for the ambiguous description and thank for the suggestion. The protein yield of full-length lanmodulin was relatively low (< 15 mg/L bacteria). To address this, the N-terminal signaling peptide was truncated to create LanM, resulting in an improved protein yield of around 35 mg/L bacteria. Based on LanM, the single-point mutated LanND was constructed, which showed a slightly increased protein yield of around 40 mg/L.

We have included the information regarding protein yields in Methods section and revised Figure 1a legend as “Lanmodulin (LanM) gene was modified to improve protein yield through truncation of signaling peptide and to enhance metal loading capacity *via* single-point mutation.”.

Are the CD spectra in Figure 1e for the apo or Gd-bound proteins? These spectra look a bit different from previously published spectra for LanM. The features at 205 and 220 nm or so do not seem distinct as they do for LanM in the literature (e.g. ref 22).

Response: The CD spectra were acquired under a cation-bound state. It’s important to note that the spectra were obtained under a buffer condition different from ref 22, where metal concentration was rigorously controlled using various chelators. In our study, we aim to assess folding properties under more physiologically relevant conditions, so the measurements were conducted in simulated physiological solutions which contain relatively high concentration of calcium (~2 mM), potassium and sodium. Since the protein binding sites are expected to be saturated by Ca²⁺, the spectra represent cation-bound proteins. For clarity, we have provided more detailed description in the legend of Figure 1e: “Note that all measurements were conducted for 50 μM proteins in a simulated physiological solution containing 135 mM NaCl, 5 mM KCl and 2 mM CaCl₂.”

It is a bit confusing that Figure 1h mentions apparent K_ds in the micromolar range. I understand that this is in the context of a competition assay with Fluo-5N, but I think this could be clarified for readers who may be less familiar with the determination of K_ds for tight binding molecules. How many binding sites on the protein did the authors assume for their K_d calculations?

Response: We have revised the presentation and description of K_ds of LanM and LanND in Figure 1h for clarity.

Although EF1, EF2, EF3 and EF4 share similar or same negatively-charged residues in LanM or LanND, their sequences and structures are not entirely identical. This makes it impractical to calculate individual K_d values for each binding site. When performing titration, we have to assume the entire protein as a whole entity to calculate the molecular apparent K_d values, just like previous report (ref.18). Therefore, in our work, the protein is assumed to possess “one binding site” for the

calculation.

Figure 1h, Determination of apparent Gd^{3+} binding affinity for LanND by plotting fluorescence intensity against protein concentration.

Figure 1k: how much Gd was bound to LanND after filtration and ICP-OES in the protein used for this experiment?

Response: To load Gd^{3+} into LanND, we mixed Gd^{3+} and proteins with a molar ratio of 6:1 to ensure complete saturation of LanND. After incubation and removal of unbound Gd^{3+} , we conducted ICP to quantify the Gd amount bound to LanND. The summarized results from multiple experimental batches revealed that approximately 67% of the introduced Gd^{3+} was bound to the proteins, yielding a calculated binding stoichiometry of approximately 4.02 for LanND (Supplementary Fig. 4).

Supplementary Fig. 4 | Assessment of metal-loading capacity *via* ICP-OES.

In the conditions in Supplementary figure 4, the wild type LanM binds roughly 4 equivalents and

the plot at left does not look very different from the LanND up to 1.5 mM protein concentration. On the plot at right, LanND binds roughly 5 equivalents Gd, but the protein would only be expected to have 4 binding sites. To me, this figure does not clearly demonstrate that LanND binds significantly more Gd than LanM under these conditions.

Response: Yes, we agree with the reviewer's comment that **Supplementary Fig. 4** doesn't directly demonstrate a higher Gd binding capability in LanND compared to LanM. Considering the challenges in fully removing weakly bound Gd^{3+} in the EF4 binding site using desalting columns and spin concentrators, it's reasonable to expect a bound Gd to LanM ratio of approximately 4. Additionally, higher concentrations of LanND may potentially introduce greater deviations in the results. Taking these factors into account, we only emphasize the incomplete removal of Gd from EF4 of LanM, which increases the potential risk associated with using LanM-Gd as a contrast agent.

In Supplementary figure 6, the proteins with Gd added are cloudy, which seems not ideal. At what concentration and in what buffer was this experiment done?

Response: We have adjusted the concentrations of LanND and LanND-Gd to be similar, between 1-2 mM, and suspended them in the identical buffer containing 135 mM NaCl, 10 mM HEPES at pH7.5. An updated image has replaced the previous one in **Supplementary Fig. 6**.

Supplementary Fig. 6 | Visual and SDS-PAGE comparison of proteins before and after Gd^{3+} ion loading.

MRI section, p. 4. Some relaxation data relevant to performance of the protein as a Gd MRI contrast agent has been reported for LanM (<https://pubs.acs.org/doi/10.1021/jacs.1c07103>). The

prior work should be cited and the authors should discuss their results in the context of the WT protein. Also, because Magnevist only has $q=1$, I wonder whether comparison in the text to an agent with $q=2$ might be more fair? (No need to do experiments with another small molecule; I think discussion in the text would be sufficient.) I also think that a more extensive comparison in the text to the Yang group's protein-based MRI contrast agent (reference 27) would be appropriate.

Response: We thank the reviewer for recommending the reference. In the reference paper, the author examined the temperature-dependent reduced transverse relaxation rate of water ^{17}O under NMR measurements (11.7 T), which is an important indicator for MRI contrast agent. However, this parameter is different from the relaxation parameters of water protons obtained by MRI, making direct comparison challenging. Nevertheless, motivated by this comment, we searched more related literature and found a preprint paper that also measured the relaxivity of Gd-bound LanM (Doi: 10.1101/2023.01.05.522788). In that study, the authors constructed a genetically-encoded LanM-based Gd^{3+} sensor (GLamouR). They tested r_1 and r_2 of GLamouR-Gd conjugate under a magnetic field of 7T, which were $6.0 \text{ mM}^{-1}\text{s}^{-1}$ and $41.85 \text{ mM}^{-1}\text{s}^{-1}$, respectively. In line with their results, we determined the r_1 and r_2 of LanND-Gd under 7T to be about $6.91 \text{ mM}^{-1}\text{s}^{-1}$ and $31.16 \text{ mM}^{-1}\text{s}^{-1}$ (**Supplementary Fig. 8-9**). The slight difference may be resulted from difference protein molecular structures or experimental errors. We have incorporated the discussions and references at corresponding position in the revised manuscript.

Regarding the suggestion for water coordination, we appreciate the comment. But to our knowledge, all clinically available agents typically exhibit only one coordinated water molecule ($q=1$). Several synthetic small molecules with $q=2$ have been reported by other researchers (<https://doi.org/10.1021/cr980440x>), but most of them only provide related parameters with NMRD, which are inappropriate to be compared with the data of LanND-Gd from MRI. Moreover, there is a lack of related imaging performance of these agents, making it hard to do the comparison. Given above situation, we chose to compare our LanND-Gd with another protein-based agent, CA1.CD2, developed from Jenny J. Yang. CA1.CD2 also features two coordinated water molecules (ref.27). We have included related discussion on CA1. CD2 and other relevant work from Yang's group in the revised manuscript.

Figure 2b and p. 5, line 132: a hydrodynamic diameter of 4.7 nm for LanND is quite high. In reference 23, the hydrodynamic diameter of LanM is reported as 3.5 nm, which is reasonable given the NMR and crystal structures. Could this mean that the LanND protein is not folded as fully as LanM?

Response: Based on the CD spectra of LanM and LanND, along with the supportive predicted LanND structure and various functional titration experiments, it's evident that LanND still preserves normal folding properties and functionalities. In order to elucidate this problem, we repeated size measurements for LanND and LanND-Gd, with LanM and LanM-Gd as control (**Figure. 2b, as shown by earlier response**). These updated data exhibited that all measurements share similar sizes, suggesting that LanND maintains the normal 3D structures as LanM.

Line 159: Can the authors clarify whether the same amount of Gd was injected in the case of the Magnevist studies as with the Gd-LanND?

Response: Yes, the injection amount of Gd is identical to that of Magnevist (both are 0.1 mmol/kg). We have further highlighted the dose in revised manuscript.

Lines 166-168: these figures should be 2h and 2i, not 1h and 1i

Response: We have corrected them in the revised manuscript.

Line 169 “demonstrating the stable circulation of LanND-Gd without detectable breakdown or leakage of Gd³⁺”: While Gd is not detected in low molecular weight fraction of the serum, it does not necessarily mean that all the Gd still in circulation is still associated with LanND. See additional information in the major comment above.

Response: Yes, that's correct. This piece of evidence may not support a confirmative conclusion of no leakage of Gd³⁺ from proteins. We have supplemented the evidence with functional titration tests of LanND under various conditions, to reinforce the claim (**Supplementary Fig.11**). Even with interference from various factors, including Ca²⁺, temperature, pH, BSA, LanND consistently

demonstrates its normal Gd^{3+} binding properties showing an effective competition concentration of around 0.4 μM . But we agree that while these findings offer indications that LanND can sustain its functionality in a complex environment, its true performance *in vivo* remains difficult to ascertain. We have rephrased the wording as: “This, together with previous indications of LanND’s remarkable stability in complex situations, demonstrate the strong likelihood of stable circulation of LanND-Gd without obvious breakdown or leakage of Gd^{3+} .”.

Renal clearance section, p. 6: this work should be discussed in the context of the work by Boros, mentioned earlier. The kidney uptake is similar to what was observed for LanM injected with radioactive La, where bone deposition was also observed. Did the authors look for uptake of Gd by bone, which would be clear indication of Gd dissociation from the protein?

Response: Thanks a lot for the very constructive comments. Just as stated in the response to the major comment earlier, LanND-Gd enhanced the protein’s binding affinity to Gd^{3+} , but little bone deposition can still be detected. It’s known that the commercially used agents, including Magnevist, also has bone deposition even after even several years. This is indeed an important problem that needs further improvement in the future, and we have highlighted this insight in discussion section.

Reviewer #2 (Remarks to the Author):

This is a *magnus opus* describing the synthesis and extensive testing of a new customized protein, lanmodulin, that binds 4 gadolinium atoms. It's a smart and timely idea. The protein outperforms a clinically widely used single gadolinium chelate (Magnevist) in terms of relaxivity (efficacy of contrast enhancement), has a "longer" blood half-life, and better anatomical details on imaging. The study is very comprehensive. The inclusion of GFP to validate the MRI with optical imaging is also very nice.

Response: We sincerely appreciate the reviewer's positive feedback.

I have a few major and minor comments that need to be addressed.

Major:

-There are some questions about blood clearance and retention, in fact I am puzzled by Fig. 2g, where the single chelate Magnevist appears to have a longer blood half life compared to Gd-lanmodulin, reversing all conclusions on an increased blood half-life of the latter. It is just said that Magnevist accumulates in the eye (I may be ignorant, but have never heard of it) but there are other tissues showing a prolonged contrast enhancement (longer than Gd-lanmodulin) in all 3 imaging planes. This cannot be ignored.

Response: We thank the reviewer for raising the ambiguity. To further validate the accumulation of Magnevist in eyes, we also conducted brain imaging under 3T. Indeed, Magnevist exhibited a brighter circular imaging pattern in the eyes compared to LanND-Gd (**Supplementary Fig. 13**). Moreover, we tested more mice under 9.4T and repeated the examination depicted in Figure.2g, the results of which were summarized in the figure attached below (**Figure. 2g and Supplementary Fig. 14**). These images confirmed that Magnevist accumulated in the eyes of all three mice, whereas LanND-Gd showed clear excretion. And this phenomenon of clinical Gd-based agents has also been reported by other researchers (Doi: 10.2463/mrms.mp.2021-0100; <https://doi.org.libproxy1.nus.edu.sg/10.1212/WNL.0000000000005123>).

Based on the results from the two additional mice, it is evident that LanND-Gd has clearly prolonged contrast enhancement performance compared to Magnevist. Previous confusion maybe resulted from animal variations, which prompt us to revise the previous figure and corresponding explanations in the revised manuscript.

Supplementary Fig. 13 | Comparison of T_1 -weighted eye images enhanced by Magnevist (a) or LanND-Gd (b) under a 3T scanner.

Supplementary Fig. 14 | Comparison of T_1 -weighted brain images enhanced by Magnevist (a) or LanND-Gd (b) under a 9.4 T scanner.

-Please calculate the blood half life from Fig 2i and provide this. Is curve fitting mono- or bioexponential? Does it have a short clearance and a long clearance component or just one. More time points may be needed for a good fit but this is a very simple and straightforward experiment.

Response: Thanks for the constructive comment. We have incorporated more time points into Fig 2i, and the curve fitting follows a monoexponential model, which indicates only one clearance component (**Figure. 2i**).

Figure 2i. Plots of relative Gd^{3+} concentration in the retentate and filtrate over time.

-How much Gd is eventually retained? This will be important long term. Collect all urine, measure Gd and subtract from the amount injected. Is 99.9% excreted at the end? Even a few % retainment may be of concern, eventually.

Response: Characterization of the long-term retention of gadolinium (Gd) is indeed crucial. To assess the retention, we collected urine samples and quantified the excreted Gd to estimate the retention percentages. Our calculations indicate that over 30% of the injected Gd was excreted within the first day. However, due to operational errors in urine collection and potential interference from the fecal route, achieving a high level of excretion, such as 99.9%, seems impractical. Therefore, we focused on examining agent retainment in major organs. Our current data revealed that only kidney still exhibited significant amount of the agents after 4 hours (**Supplementary Fig. 17**). Subsequent measurements of the retainment in kidney on day1, day3 and day7, demonstrated that less than 1% retention in kidneys after one week (**Supplementary Fig. 17**). Certainly, experimental errors are inherent, conducting future studies with a larger sample size

to analyze urine and feces may facilitate a more comprehensive investigation. And we have included discussion in this regard in our revised manuscript.

Supplementary Fig. 17 | Distribution of Gd³⁺ in various organs after injection of LanND-Gd.

-All potential toxicity tests are performed very quickly after injection (1 week). Possible side effects will manifest themselves much later, esp in the CNS. It is understood that long-term follow up studies are beyond the scope of this work, but there should be a clear limitation statement that the behavioral and toxicity tests may tell us not as much as the authors claim.

Response: Yes, that’s correct. In our animal behavior assessments, we sequentially injected two doses of the agents with a one-week interval, which ensures a sufficient time to induce immune responses, and subsequently evaluated the behaviors after around two weeks. However, even though these data provide some insights, it remains a relatively short timeframe for observing long-term toxicity effects. To address this, we conducted an examination of brain functions by recording neural firing, a more sensitive indicator, in the prefrontal cortex, a region closely related to emotion and memory. The results also indicated no significant differences compared to the control group (**Figure. 4m-o**). Nevertheless, long-term toxicity concerns warrant more systematic investigation. We have revised our claim in a more rigorous manner and included a discussion in this regard as follows: “The toxicity evaluation reveals no acute side effects associated with LanND-Gd. However, further systematic investigation is certainly imperative to assess long-term safety issues and potential Gd deposition in bones or skin before considering clinical translations.”.

-Stability/dechelation/ K_d max studies in plasma would also be valuable.

Response: Yes, this is indeed a valuable suggestion. Initially, we performed titration in pure plasma, but we found that the fluorescence of Fluo-5N (working concentration: 1 μM , in 96-well plate) was saturated because of the high Ca^{2+} concentration in plasma ($\sim 2 \text{ mM}$) (**Supplementary Fig. 11a**). Through experimental calculation, the binding affinity of Fluo-5N to Ca^{2+} is around 90 μM (**Supplementary Fig. 11b**). Given the impracticality of using very high concentrations of Fluo-5N, such as 1 mM, in titration, we investigated the influence of different factors, including Ca^{2+} , pH, temperature, BSA, respectively (**Supplementary Fig. 11c-f**). None of these factors showed obvious effects on the protein's competitive ability to bind Gd^{3+} .

Supplementary Fig. 11 | Investigation of Gd^{3+} binding affinity for LanND under various interference factors.

Finally, considering that concentration of Gd^{3+} used in animal imaging is around 1 mM, which is close to the concentration of plasma Ca^{2+} , it is reasonable for us to perform the titration under diluted plasma conditions to reduce Ca^{2+} to a concentration similar to that of Gd^{3+} (1 μM for the titration system, **Supplementary Fig. 11f**). The diluted plasma avoids the dye saturation problems caused by high Ca^{2+} content. And the results also exhibited that plasma had no significant influence

on LanND's binding affinity to Gd^{3+} , suggesting that LanND can maintain its normal functionality even under complicated situations.

Minor:

-Line 119: add “molecular” between effective and relaxivity for clarity.

Response: We have adopted the advice for the more precise expression of “effective molecular relaxivity”.

-Line 122: inverse relationship of r_1 and r_2 . This is not “Interestingly”, much literature on T1 NMRD profiles exist since the mid 1980s. PMID: 11902904 describes a study on a Gd-protein (HSA) and the r_1/r_2 relationship on field dependence.

Response: Yes, that is correct. We intended to highlight that the inverse relationship of LanND-Gd is much steeper compared to that of Magnevist, but we overlooked comparing this property with other protein-related agents. In the revised version, we have replaced the phrase “Interestingly” with “Notably”.

-Line 174. “nanoparticles circulate excessively”. What is meant with “excessively”? This suggests extensively, or a long blood half life? But they have a very short half live before taken up by the RES, much shorter than Gd-lanmodulin. Please remove or rephrase.

Response: Sorry for the lack of clarity. Here, “excessively” does not refer to blood half-life, but rather to “very long retention time in animals, potentially even permanent retention in animal body”. We have rephrased the expression as follows: “Unlike conventional nanoparticle-based contrast agents, which are prone to endocytosis by cells or tissues during circulation in the bloodstream and subsequent accumulation in organs, leading to concerns about potential side effects or toxicity, the compact size of LanND-Gd (< 5 nm) falls within the range of complete urinary excretion from the body.”

-Line 184: “Accumulation” in kidneys. This suggests uptake. It merely passes through.

“Distribution” may be a better term.

Response: We have replaced the term “accumulated” with “distributed”.

-Line 196: This protein is not a nanoparticle and the lung uptake is unexpected. Needs further thought. But even nanoparticles do not accumulate in lungs. Mostly liver, spleen, and lymph nodes.

Response: Sorry for not expressing properly. Following intravenous injection of the protein agent, it would be rapidly transported through the venous system to the heart, and then enters the lungs *via* the pulmonary artery along with the blood flow. This occurs because the right ventricle of the heart pumps blood to the lungs for gas exchange. Consequently, the majority of intravenously injected drugs initially reach the lungs before being distributed to various tissues and organs throughout the body *via* the arterial system. Therefore, it is not uptake, it just passes the lungs. A recent study conducted single-particle tracking *via* PET, and they observed that the nanoparticles tend to stay in the lungs after intravenous injection (<https://doi.org/10.1038/s41565-023-01589-8>). While the protein agent is more accurately classified as a drug rather than a nanoparticle, the distribution patterns of nanoparticles can offer some valuable insights. We have rephrased our wording as follows: “Similar to most intravenously administered drugs, the agent initially distributed temporarily in the lungs along with the blood flow and diminished rapidly within 30 min.”.

-Line 201. The kidney itself is not the primary route of excretion. That is the bowel, where the term “excrements” has been coined from.

Response: Yes, that’s right. We have corrected the inaccurate wording by removing the emphasis on “primary route” in the revised manuscript.

-Line 203 and 216: It is (dys)function, not (dys)functions.

Response: We have corrected them in the revised manuscript.

-Figure 3e. Please provide abbreviations indicating organs. One has to sort of distill it from panel 3f.

Response: The organs have been appropriately labelled at their corresponding positions.

Figure. 3e, Representative distribution of GFP-tagged protein agents in major organs.

Reviewer #3 (Remarks to the Author):

Thank you for asking me to review this manuscript in which a new MRI contrast agent is described, based on a protein (LanM) that is modified with a single point mutation and is then saturated with gadolinium. Results in terms of imaging show that there is a reduction in relaxation times, resulting in brighter images from different organs and better resolution of the vasculature as compared to standard gadolinium contrast (Magnevist). I am reviewing this from the point of view of a clinician; it will be important to ensure that there is sufficient review of the MRI physics aspects.

Response: We appreciate the reviewer's valuable comments.

The main potential weakness relates to the renal elimination of the new contrast agent, and that all of the experiments were performed in animals with normal renal function (in particular toxicity studies). One of the main limitations of standard gadolinium (in particular in research settings) is its use in patients with impaired kidney function – in contrast the use of gadolinium in patients with normal kidney function is generally viewed as much lower risk. It is possible therefore that due to reduced clearance, the toxicity profile may be substantially different/worse in those with impaired renal function, and this new agent may end up having exactly the same issues as standard gadolinium.

Response: We thank the reviewer for the invaluable comments from a clinical perspective. It's indeed true that currently used Gd-based MRI agents may exacerbate kidney injuries in patients with renal diseases. In our work, we aim to utilize protein-based agents to offer an alternative due to their better biocompatibility and higher relaxivity. Through our research, we observed that these agents can be effectively excreted out of the animal body, even in case of kidney injury, after several hours. We fully agree with the reviewer that potential toxicities on injured kidneys requires further systematic evaluation in the future, particularly if we intend to employ this type of agent as an option for the patients with renal injuries. For example, given the significantly superior relaxivity performance of LanND-Gd, the administration dose of this agent may be substantially reduced, which could further mitigate potential concerns. We have expanded our discussions on this point

in the revised manuscript as follows:” It’s noteworthy that currently used Gd-based MRI agents are unsuitable for patients with kidney diseases as they may worsen kidney injuries. LanND-Gd may present a potential alternative, offering better biocompatibility and higher relaxivity. Additionally, this agent could help substantially reduce the required dosage to further mitigate potential effects on kidney function.”

In the introduction, it is not clear what the aim of the new contrast agent is – shortening relaxation times, visualization of vasculature and evaluation of kidney function are all mentioned, but more precision about the aims of developing this specific contrast agent would be helpful.

Response: Our aim is to overcome the limitations associated with clinical MRI agents characterized by low relaxivity and short time-window for imaging. We intend to develop a new generation of agents to expand the applications of MRI techniques to fine structural and organ functional imaging. In this context, we showcase brain imaging and kidney monitoring as examples to underscore the advantages of our new agent. These demonstrations also serve as a potential reference for future clinically relevant applications. In the revised introduction, we prioritize the development of the new agent over disease-specific details to highlight the core aim of this study.

There is a lack of justification of how and why LanM was chosen, and this should be included in the introduction.

Response: Currently, all of the clinical contrast agents are limited by their ability to coordinate only one water molecule, which largely restricts their relaxivity outcomes. In contrast, LanM has a unique ability to coordinate two water molecules, suggesting a potential for superior relaxation performance and improved *in vivo* imaging outcomes. We have emphasized this point in the introduction section as follows: “The discovery of a high-affinity lanthanide-binding protein, lanmodulin (LanM), has inspired us to develop a high-performance MRI contrast agent owing to its ability to coordinate two water molecules. This unique property holds the potential to enhance the relaxivity outcome compared to commonly used clinical agents, which typically only coordinate one water molecule. Hence, we have engineered a LanM-derived protein MRI contrast

agent tailored for high-resolution vascular imaging and effective monitoring of kidney dysfunction (Fig. 1a).”

It is possible that gadolinium dissociates from LanM after infusion, so it was good that *ex vivo* experiments were included that suggested that this was less likely.

Response: Thanks for the feedback. In the revised version, we also incorporated additional evidence, including protein titrations under various conditions, to reinforce that LanND can sustain normal functionality in simulated *in vivo* situations.

Suppl Fig 13 – although the results are reported as non-significant, there are some concerning patterns seen in relation to potential toxicity. Somewhat offset by toxicity studies based on histology although these will not capture longer term or functional effects. This should be discussed and may need further studies to reassure on this point.

Response: Thanks for raising potential concerns regarding safety issues. To address these concerns more convincingly, we conducted animal behavior tests with a larger sample size to minimize the impact of experimental variation (**Supplementary Fig. 19**). The results indicated that no abnormal neural behaviors were observed. Following behavior evaluation, we further examined the neural firing properties of the prefrontal cortex, which is associated with emotion and memory, to better characterize any potential neural effects (**Figure. 4m, n, o**). We found no significant difference between the control group and the group injected with LanND-Gd, further suggesting the absence of acute toxicity issues associated with LanND-Gd.

Certainly, all the tests were conducted over a relatively short term, within a two-week period. Therefore, we can only conclude that low acute toxicity was observed and long-term effects needs further assessment in the future. We have adjusted our expressions in the revised manuscript.

Supplementary Fig. 19 | Emotional test for potential immunogenic damage.

Figure.4. m, Representative traces of action potential firing in prefrontal cortex neurons from control or LanND-Gd injected group. **n**, Statistical analysis of neural firing frequency induced by varying injection currents. **o**, Summary of rheobase of the neurons.

REVIEWER COMMENTS (Manuscript # NCOMMS-24-06716A)

Reviewer #2 (Remarks to the Author):

Thank you for making the many revisions. Coming back to my previous point, I am still confused about previous Fig 2g and new Suppl Fig 14. Outside of the eye, other tissues clearly have a longer residual (10,20,30 min) diffuse contrast for Magnevist compared LanND-Gd, contradicting the said longer blood-half life of the latter. This applies to all 3 mice. Throughout the manuscript, it is stated at different places that LanND-Gd has a longer retention (which seems to have been interpreted here as having a longer blood half life). “Retention” normally refers to the residual cell/tissue amount after having been cleared from blood

This agrees with the very short half life of LanND-Gd that has now been calculated (thank you) from Fig 2i as being 6 minutes. This is much shorter than that of Magnevist (which has a biexponential fit with 12 and 96 min half-life values in humans). Magnevist clearance data for mice are not given here (could do the same expt as Fig 2i). What is this paradox?

Response: We sincerely appreciate the reviewer’s feedback and the opportunity to further improve our manuscript. We have addressed the concerns in the following three aspects:

1. Clarification on the terminology of “retention” vs. “blood half-life”

We acknowledge the confusion between “retention” and “blood half-life”. In the manuscript, we initially used "retention" to describe the persistence of LanND-Gd in kidneys after blood clearance, which may have led to misunderstandings. We have revised our manuscript to use “renal retention” to specifically refer to the residual levels in the kidneys following clearance from the bloodstream.

2. Comparison of residual contrasts in non-ocular tissues (Fig. 2g and Suppl Fig. 14)

Regarding the comment that Magnevist shows longer residual diffuse contrast in non-ocular tissues compared to LanND-Gd at the 10, 20, and 30-minute time points, we hypothesized that this difference may be attributed to their distinct chemical properties. The smaller molecular size of Magnevist likely allows it to diffuse effectively into tissue interstitial spaces, whereas the larger size of LanND-Gd might limit its tissue infiltration, leading to faster clearance from the bloodstream.

However, in the kidneys, LanND-Gd demonstrates longer retention than Magnevist (**Fig. 3h**). This is likely due to the fine structure of the kidney (**Fig. 3i**), where larger molecules take longer to exclude. This property provides an advantageous time window for kidney imaging with LanND-Gd, which we emphasized in our manuscript.

3. Pharmacokinetics of Magnevist in mice

We appreciate the reviewer’s suggestion to include pharmacokinetic data for Magnevist in mice

for comparison. While the pharmacokinetics of Magnevist in humans is well-established, drug clearance rates can vary across species. To address this, we conducted the same experiments as outlined in **Fig. 2i** for Magnevist, enabling direct comparisons with LanND-Gd.

Fig. 2i., Plots of relative Gd^{3+} concentration in the retentate and filtrate, as well as in bones, over time. The pharmacokinetics in the bloodstream for LanND-Gd was fitted using a monoexponential function ($\tau = 5.99 \pm 1.65$), with Magnevist used for parallel comparison ($\tau = 7.12 \pm 0.66$). Note that from 2 h to 24 h, ICP-OES indicated “not detected” as the Gd^{3+} content fell below the limit of detection, so their values were set as “0”. Data were shown as “Mean \pm S.E.M.”, three to five mice were analyzed for each time point.

Magnevist was injected into mice at the same dosage as LanND-Gd, with blood samples collected at identical time points. Gd levels at each time point were analyzed using ICP-OES. To minimize potential harm to animals, blood samples were only taken starting 5 min post-injection. By fitting the data to an exponential decay function, we found that a single-index exponential model provides a good fit, suggesting that the initial distribution phase is likely completed within the first 5 min.

For the second phase of elimination, which is the primary focus of our study, the half-life was approximately 7 min, comparable to that of LanND-Gd (6 min). This duration is significantly shorter than that in humans (about 96 min), which may be attributed to several factors, including a faster heart rate (about 300-700 beats/min in mice vs. 60-100 beats/min in humans), increased blood flow, a higher metabolic rate, more active enzyme activity, and quicker excretion.

In summary, LanND-Gd and Magnevist exhibited similar elimination profiles in bloodstream, but Magnevist showed a more pronounced residual presence in the eyes and brain tissues. Conversely, LanND-Gd was cleared from the kidneys more slowly than Magnevist, providing a sufficient time window for kidney imaging. Importantly, the protein nature of LanND-Gd allows for easy modification with peptide tags, such as proline-rich peptide sequence, XTEN peptides, Fc fusion, CEX peptides in GLP-1 analogs, to extend its circulation time for specific imaging scenarios in the future, such as in blood vessel imaging.

We have revised our manuscript to replace “prolonged retention” with “prolonged renal retention” to prevent any potential confusion. We sincerely thank the reviewer once again for the valuable feedback that has helped to improve the technical content of our work.

Reviewer #3 (Remarks to the Author):

Thank you - all of my points have been addressed

Response: We thank the reviewer for the positive feedback.

Reviewer #4 (Remarks to the Author):

As a substitute reviewer, I find that the manuscript presents extensive work on the modification of lanmodulin for use as an MRI contrast agent. The results are impressive, and the effects of a single point mutation are fascinating. The reviewers' questions were well thought out and demonstrated a deep engagement with the manuscript.

I would have appreciated more characterization of the mutation effects. For instance, the changes in the affinity of the EF-hand compared to LanM could have been assessed through mutations aimed at reducing the affinity of the highly affine sites 1 to 3, which would have simplified the data complexity. However, I acknowledge the breadth of work completed and recognize that the focus was more on the application in the organism.

I believe the review has helped improve the quality of the manuscript. The authors have addressed all comments appropriately, responding with relevant experiments and clearer textual representations. I have no further objections to the publication.

Response: We thank the reviewer for the positive feedback on our manuscript. We are glad that the reviewer found our work on lanmodulin modification for MRI contrast agents impressive and appreciated the impact of the single-point mutation.

We also acknowledge the reviewer's suggestion to further characterize the effects of mutation, particularly by reducing the affinity at EF-hand sites 1 to 3. While these tests are beyond the scope of our current study, we find them highly interesting and plan to pursue them in future research. We appreciate your understanding in this regard.

We are pleased that our revisions and responses have addressed the reviewer's concerns and that the reviewer has no further questions. We sincerely thank the reviewer once again for the valuable input.